# Novel cell types and developmental lineages revealed by single-cell RNA-seq analysis of the mouse crista ampullaris

**Brent A Wilkerson[1,2†], Heather L Zebroski[1,2], Connor R Finkbeiner[1,2], Alex D Chitsazan[1,2,3‡], Kylie E Beach[1,2], Nilasha Sen[1], Renee C Zhang[1], Olivia Bermingham-McDonogh[1,2*]**

[1]Department of Biological Structure, University of Washington, Seattle, United States; [2]Institute for Stem Cells and Regenerative Medicine, University of Washington, Seattle, United States; [3]Department of Biochemistry, University of Washington, Seattle, United States

**Abstract** This study provides transcriptomic characterization of the cells of the crista ampullaris, sensory structures at the base of the semicircular canals that are critical for vestibular function. We performed single-cell RNA-seq on ampullae microdissected from E16, E18, P3, and P7 mice. Cluster analysis identified the hair cells, support cells and glia of the crista as well as dark cells and other nonsensory epithelial cells of the ampulla, mesenchymal cells, vascular cells, macrophages, and melanocytes. Cluster-specific expression of genes predicted their spatially restricted domains of gene expression in the crista and ampulla. Analysis of cellular proportions across developmental time showed dynamics in cellular composition. The new cell types revealed by single-cell RNA-seq could be important for understanding crista function and the markers identified in this study will enable the examination of their dynamics during development and disease.

**\*For correspondence:**
oliviab@uw.edu

**Present address:** †Department of Otolaryngology-Head and Neck Surgery, Medical University of South Carolina, Charleston, United States; ‡CEDAR, OHSU Knight Cancer Institute, School of Medicine, Portland, United States

**Competing interests:** The authors declare that no competing interests exist.

## Introduction

The vertebrate inner ear contains mechanosensory organs that sense sound and balance. In mammals, the cochlea senses sound, the saccule and utricle sense horizontal and vertical acceleration, respectively, and the cristae in each of the semicircular canals sense angular head movements, which is critical for maintaining balance and the vestibulo-ocular reflex (*Highstein and Holstein, 2012*). Cristae exhibit significant age-related degeneration of hair cells, particularly type I hair cells (*Rauch et al., 2001*; *Lopez et al., 2005*). Dysfunction of crista is therefore implicated in balance disorders and falls in the elderly, as well as in the pathology of Meniere's, benign paroxysmal positional vertigo and other balance disorders.

Whereas other sensory organs in the inner ear have been profiled by single-cell RNA-seq (scRNA-seq), no studies to date have analyzed the crista ampullaris. Analysis of scRNA-seq data from the utricle, cochlea, and endolymphatic sac have revealed developmental transitions and rare cell types of possible importance to the etiology of hearing and vestibular disorders (*Burns et al., 2015*; *Honda et al., 2017*; *Yamashita et al., 2018*; *Korrapati et al., 2019*; *Hoa et al., 2020*; *Ranum et al., 2019*; *Petitpré et al., 2018*; *Shrestha et al., 2018*; *Sun et al., 2018*). The sensory epithelium of the crista includes type I and type II hair cells each having distinct morphology and synapses, support cells and glia (*Desai et al., 2005*). Five nonsensory cell types of the ampulla have been described morphologically: (1) transitional epithelial cells bordering the sensory epithelium, (2) periotic mesenchyme surrounding the ampulla epithelium, (3) dark cells of the ampulla, the osmiophilic and endolymph-producing cells, (4) melanocytes underlying the dark cells and (5) light cells, the osmiophobic cells of the planum semilunatum and the ampulla roof and wall (*Köppl et al., 2018*; *Kimura et al.,*

*1964*; *McLamb and Park, 1992*; *Tachibana et al., 1987*; *Nakai and Hilding, 1968*; *Bairati and Iurato, 1960*; *Ten Cate and Rarey, 1992*). Understanding the cellular composition of the crista and the gene expression in each cell type is a critical step towards understanding inner ear development, function, and vestibulopathies.

In mammals, regeneration in the vestibular organs is slow, highly variable and limited to Type II-like hair cells with immature hair bundle morphology (*Kinoshita et al., 2019*; *Sayyid et al., 2019*; *Forge et al., 1993*). An unexplained phenomenon of importance to the goal of therapeutic regeneration of hair cells in the vestibular system is the perinatal window of greater regeneration-competence in support cells (*Slowik and Bermingham-McDonogh, 2013a*). In adult cristae, a subset of support cells maintains competence to convert into hair cells in response to Notch inhibition (*Slowik and Bermingham-McDonogh, 2013b*), which implies heterogeneity of support cells in the crista.

To gain insights into the cellular composition of the crista during the perinatal window of greater regeneration-competence of support cells, we performed single-cell RNA-seq in crista ampullaris microdissected from E16, E18, P3, and P7 mice. We identified and validated many novel cell types, cell-specific markers, and developmental changes in cellular composition of the crista that are in addition to the support cells we initially aimed to study. Integrated analysis of the data provides novel insights into the pathways and transcription factors that may regulate cell-specific gene expression and functions in the crista as well as the cell types involved in vestibular disease.

## Results

### Cellular diversity in the crista ampullaris

Cells were dissociated from crista ampullaris of mice at E16, E18, P3, and P7 and their gene expression was analyzed by scRNA-seq (*Figure 1a*). Dimensional reduction was performed using UMAP to visualize transcriptional differences in cells (*Figure 1b*). Based on enrichment for known markers (*Figure 1c and d*), we identified major cell types as shown in *Figure 1b*. Clustering detects the major cell types at all four ages examined, although the proportions of some types change between E16 and P7. Cluster analysis identifies distinct cell groups enriched with known markers for hair cells, support cells, dark cells of the crista as well as nonsensory epithelial cells of the ampulla roof and wall, glia, otic mesenchymal cells, macrophages, and vascular cells (*Figure 1b–d*). Clustering also identifies transcriptionally distinct subtypes of cells. Analysis of the cellular subtypes, markers and dynamics for each major cluster is shown separately in *Figures 2*, *3*, *4*, *5*, *6* and *7* and described below. The specific tissue for dissociation into single cells included the SE, transitional epithelium, the epithelial cells making up the walls of the ampulla, ampullar associated mesenchyme and the glial cells associated with the peripheral processes of the vestibular neurons. The cell bodies of the neurons were not included.

### Heterogeneity and developmental changes in the sensory epithelium and glia

Clustering identifies six transcriptionally distinct clusters of support cells and hair cells (*Figure 2a–c*). Two clusters of hair cells show specific expression for known markers of vestibular type I and type II hair cell subtypes, *Ocm* (*Simmons et al., 2010*) and *Anxa4* (*McInturff et al., 2018*), respectively (*Figure 2c and g*). Similarly, Ocm IF in P3 B6 crista identifies cells in the hair cell layer having calyx-type synapse and lacking Sox2⁻ nuclei (*Figure 2—figure supplement 1*). Clustering also identified two support cell clusters that both express the known support cell markers *Zpld1* (*Vijayakumar et al., 2019*) and *Otog* (*El-Amraoui et al., 2001*; *Figure 2b–c*); however, the cluster-specific expression of unique markers such as *Id1* indicates heterogeneity in the support cell population (*Figure 2c and g*). The cells we label 'SC–HC transition' express a combination of support cell markers and hair cell markers (*Figure 2c*). The cells we label "NS–SC transition express some support cell markers such as *Zpld1* but not others such as *Otog* (*Figure 2c*). RNA ISH data from the Allen Developing Mouse Brain Atlas (*Miller et al., 2014*) independently validates the localization of several markers in support cells and hair cells of the E15.5 B6 crista (*Figure 2—figure supplement 2*).

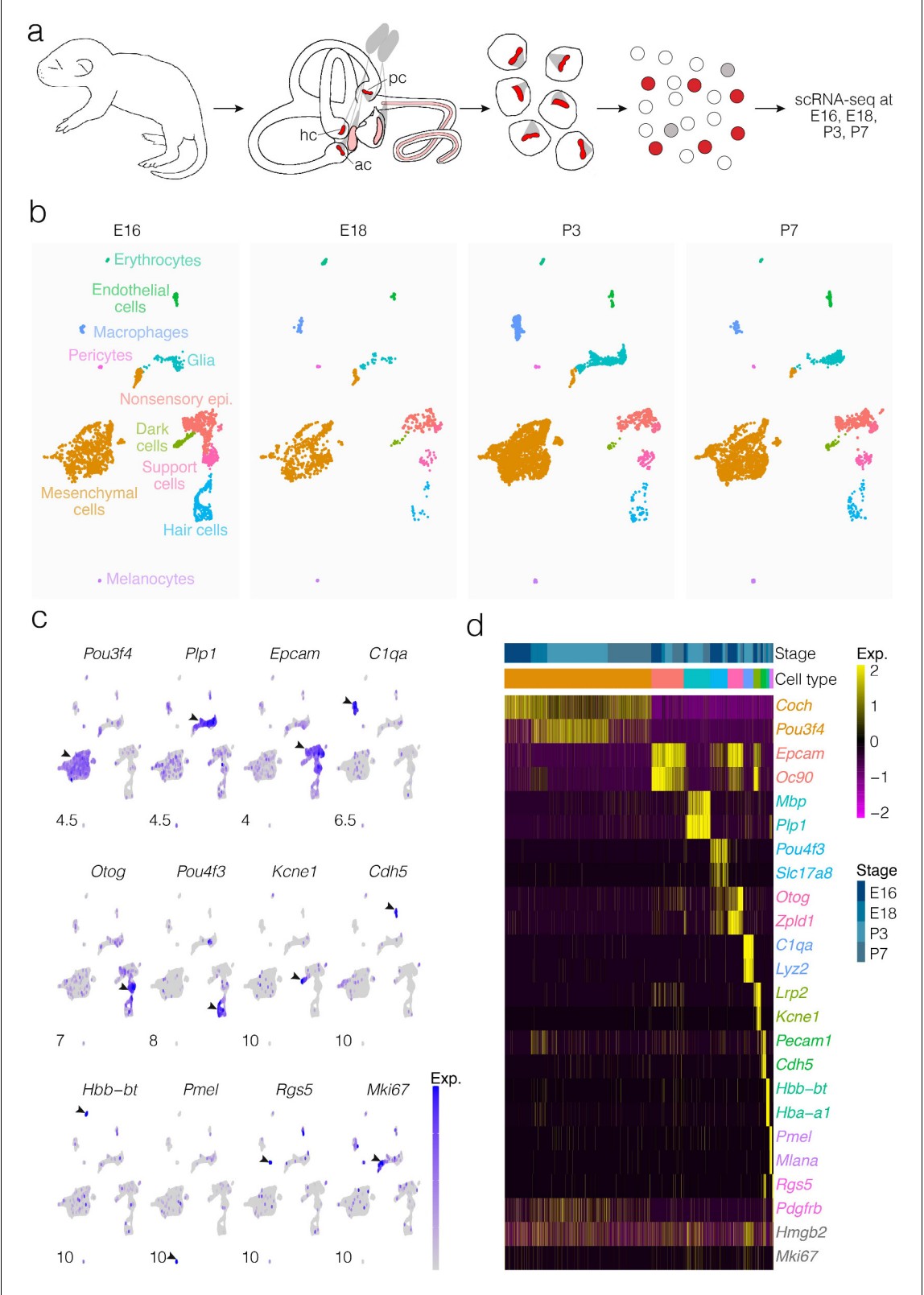

**Figure 1.** Cellular diversity in the crista ampullaris. (a) Cristae, including the ampullae, were dissected from E16, E18, P3, and P7 mice, then dissociated for single-cell RNA-seq (scRNA-seq) analysis of cell types and gene expression. (b) UMAP of batch-corrected scRNA-seq datasets from E16, E18, P3, and P7 crista ampullaris. (c) Expression of known marker genes in each of the major cell groups detected by scRNA-seq and the G2/M maker *Mki67*. (d) Heatmap showing additional known markers. *ac*, anterior crista; *hc*, horizontal crista; *pc*, posterior crista.

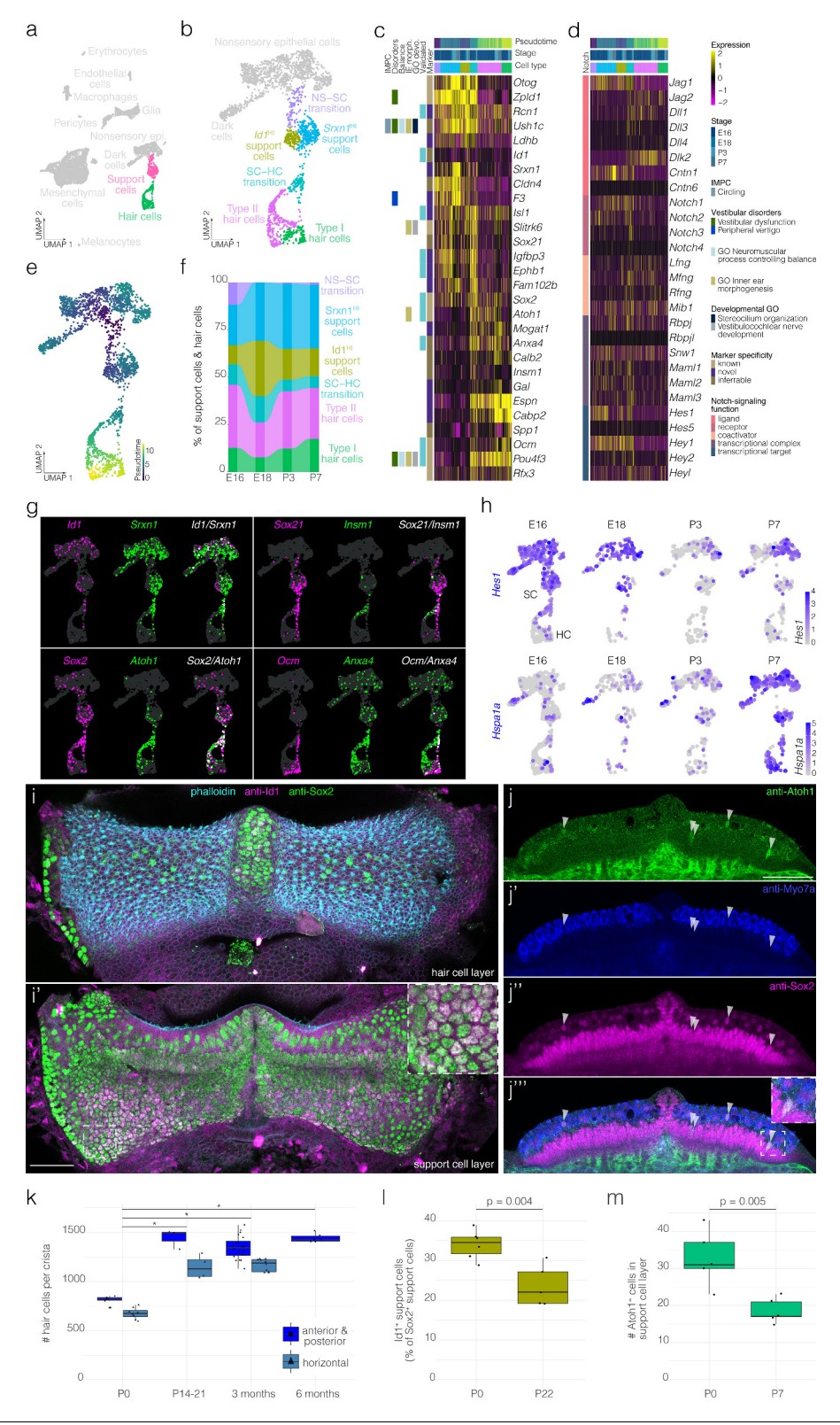

**Figure 2.** Hair cell and support cell subtypes and dynamics during development. (a) Highlights the position of support cells and hair cells in UMAP space relative to the whole dataset (i.e. E16, E18, P3, and P7 combined). (b) Cluster analysis in support cells and hair cells. (c and d) show expression of markers and Notch pathway genes, respectively, relative to cell cluster, pseudotime and developmental time. Additional color bars indicate whether markers are known or novel in specificity, implicated in vestibular diseases and dysfunction or in the Notch pathway. (e) Pseudotime of the support cell–

*Figure 2 continued on next page*

*Figure 2 continued*

hair cells trajectory in cristae. Note that the trajectory between support cells and hair cells is continuous. (f) The relative contributions of support cell and hair cell subtypes to the composition of the support cell and hair cell clusters at the indicated developmental stages. Note that the contribution of the Id1$^{HI}$ support cell subtype decreases as that of Srxn1$^{HI}$ support cell subtype and hair cells increases. (g) 'Blendplots' show the overlap of marker specificity in UMAP. (h) shows differential expression of the Notch target *Hes1* and the Hsp70 component *Hspa1a* across developmental stages in the support cell clusters. (i) Id1 (*magenta*) localization in a subset of Sox2$^+$ (*green*) support cells in P0 B6 crista relative to phalloidin (*cyan*). (j) Atoh1 (*green*), Myo7a (*blue*) and Sox2 (*green*) IF in optical sections of P0 B6 mouse crista. *Arrowheads* indicate Atoh1$^+$/Sox2$^+$ nuclei in the support cell layer. (k) Hair cell counts made manually in whole B6 cristae at P0 (n = 15), P14-21 (n = 7), 3 months (n = 30), and 6 months (n = 6). Note the addition of ~500 hair cells. *Asterisks* indicate p<1×10$^{-7}$. (l) The Id1$^+$ percentage of Sox2$^+$ support cells at P0 (n = 6) and P22 (n = 5). (m) The number Atoh1$^+$ cells in the Sox2$^+$ support cell layer per crista at P0 (n = 5) and P7 (n = 5). Scale bars = 50 μm.

The online version of this article includes the following figure supplement(s) for figure 2:

**Figure supplement 1.** Oncomodulin immunofluorescence in type I hair cells of the crista ampullaris.
**Figure supplement 2.** Localization of hair cell and support cell markers.
**Figure supplement 3.** Developmental stage and pseudotime in crista epithelial cells.
**Figure supplement 4.** RNA velocity analysis in crista epithelial cells.
**Figure supplement 5.** Localization of anti-Id1 in crista.
**Figure supplement 6.** Heat-shock pathway dynamics during support cell development.
**Figure supplement 7.** Fgf pathway in the developing crista ampullaris.

Trajectory analysis shows that the SC–HC transition cells are transcriptionally most similar to the type II hair cells (*Figure 2c–e*, *Figure 2—figure supplement 3*). Genes involved in hair cell differentiation such as *Atoh1*, *Sox21*, and *Insm1* show developmentally dynamic expression along the trajectory (*Figure 2g*). Consistent with this, RNA velocity analysis indicates a region of high velocity in the SC–HC transition that is largely in the direction of Type II hair cells (*Figure 2—figure supplement 4*). Additionally, proportions of the hair cell and support cell clusters change between E18 and P7 (*Figure 2f*). For example, hair cells and Srxn1$^{HI}$ support cells increase as a percentage of total cells in the crista at P7 relative to E18, while Id1$^{HI}$ support cells and transitional cells decrease (*Figure 2f*). Taken together, we surmised that developmental dynamics in the postnatal crista likely include hair cell recruitment from the immature support cell layer and corresponding changes in support cell composition.

To determine whether support cell clusters indeed represent distinct cell populations, we examined the localization of the cluster-specific markers Id1, Sox2, and Atoh1 in situ. Anti-Id1 IF demonstrates 20–40% of support cells in the crista (*Figure 2i and l*, *Figure 2—figure supplement 5*, *Video 1*, *Video 2*). Atoh1 IF detects ~30 cells in the support cell layer in addition to some Myo7a$^+$ and Myo7a$^-$ cells in the hair cell layer in the P0 crista (see *arrowheads* in *Figure 2j*, counts in *Figure 2m*). Similar to the cellular dynamics shown by scRNA-seq (*Figure 2f*), manual cell counts in IF labeled cristae show the addition of ~500 hair cells by P14 (*Figure 2k*),~40% decrease in Id1$^+$ support cells by P22 (*Figure 2l*) and ~50% decrease in Atoh1$^+$ cells in the support cell layer by P7 relative to P0 (*Figure 2m*).

Based on the known role of Notch-signaling in crista sensory development (*Kiernan et al., 2001*), expression of ligands, receptors and targets of the Notch-signaling pathway was of particular interest. We found enrichment of the Notch targets *Hes1*, *Hey1*, *Jag1*, *Igfbp3*, and *Slitrk6* in the support cells and the SC–HC transition cells and decline in Notch targets in maturing hair cells (*Figure 2c,d and h*). Additionally, *Hes1* and *Jag1* decline in support cells by P3 (*Figure 2d and h*). To identify novel pathways and transcription factor activity associated with support cell maturation, we performed GSEA on transcriptional changes in support cells across developmental stages. By P7, support cells show increased enrichment for heat stress response pathway and the transcriptional targets of Hsf1/2 (*Figure 2h*, *Figure 2—figure supplement 6a*). Similarly, IF for the inducible isoform of Hsp70 increases in cristae by P7 (compare in *Figure 2—figure supplement 6c and d*). Hsp70 localizes to an apical structure extending from some support cells adjacent to hair cell stereocilia bundles (*Figure 2—figure supplement 6b*).

The FGF pathway has been shown by our lab and others to be important in the development of the sensory epithelium in the cochlea (*Hayashi et al., 2008*; *Yang et al., 2019*; *Chang et al., 2004*; *Pirvola et al., 2000*) and is implicated in crista development (*Chang et al., 2004*; *Pirvola et al., 2000*) but the Fgf ligands involved in canal and crista formation are unclear. We therefore looked at

the expression of all 4 FGF receptors and all ligands in our single cell combined dataset. We find that Fgfs 1, 3, 7, 8, 9, 10, 16, and 21 are expressed in either the support cells, SC–HC transition cells or the hair cells (*Figure 2—figure supplement 7a and b*). In the crista, *Fgf21* marks the precursors of Type II hair cells, whereas *Fgf7* marks the transition to the Type I hair cells and is also expressed by a subset of these cells. *Fgf8* shows greater expression in Type I hair cells than Type II hair cells (*Figure 2—figure supplement 7a*), similar to the pattern of Fgf8-CreER activity (*Ratzan et al., 2020*), and increasing expression along the SC–Type I transition. *Fgfr1* is the predominant receptor with only sparse expression of *Fgfr2* and no expression of *Fgfr3* or *Fgfr4* within the sensory/prosensory cells. Nonsensory epithelial cells express *Fgfr2* (*Figure 2—figure supplement 7a–b*), as previously described (*Pirvola et al., 2000*), as well as *Fgfr1*. *Fgfr4* is expressed in the mesenchyme cells (*Figure 2—figure supplement 7b*), as previously described (*Hayashi et al., 2010*).

Clustering identifies four clusters of glia: glial progenitors, $Mbp^{LO}$ cells, $Npy^{HI}$ glial cells and $Mbp^{HI}$ Schwann cells. Schwann cells in the crista express known myelinating glia markers including *Mbp*, *Mpz* and *Egr2* (*Figure 3a–c*), whereas glial progenitors express G2/M markers including *Top2a* and *Mki67* (*Figure 3c*). Trajectory analysis infers a developmental lineage comprising proliferating glial progenitors, $Mbp^{LO}$ cells having intermediate levels of markers for myelinating glia and Schwann cells expressing high levels of *Mbp* and other myelinating glia markers (*Figure 3b–d*).

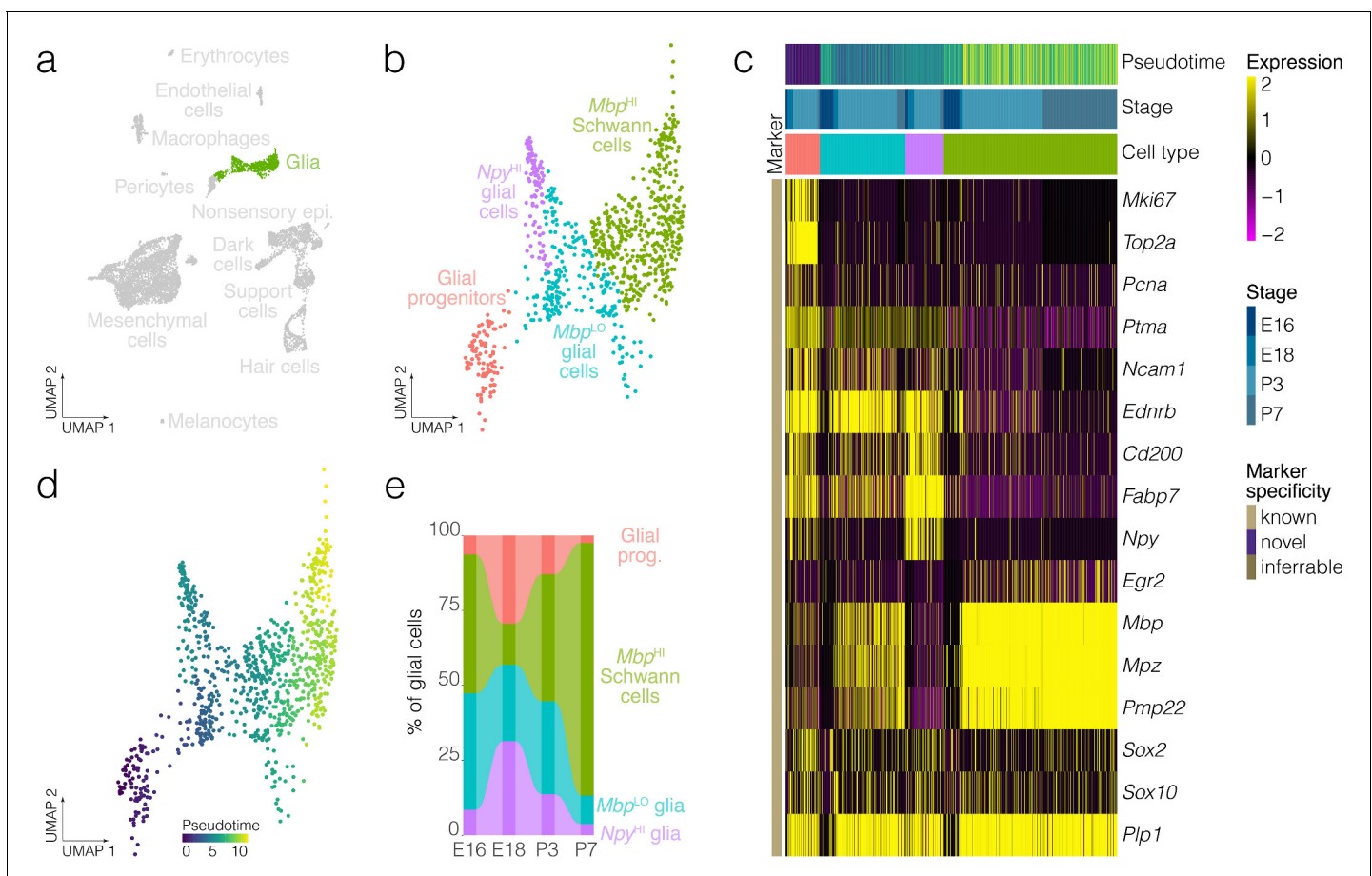

**Figure 3.** Glial cell diversity and dynamics. (**a**) Highlights the position of glial cells in UMAP space relative to the whole dataset. (**b**) Cluster analysis in glia. (**c**) Expression of markers relative to cell cluster, developmental stage and pseudotime. (**d**) Pseudotime of the developmental trajectory in glia. (**e**) The relative contributions of cell clusters to the composition of glia at the indicated developmental stages. Note that Schwann cells and myelinating cell markers increase with the developmental time.

The online version of this article includes the following figure supplement(s) for figure 3:

**Figure supplement 1.** Developmental stage relative to UMAP in crista glial cells.

**Figure supplement 2.** Localization of glial cell markers.

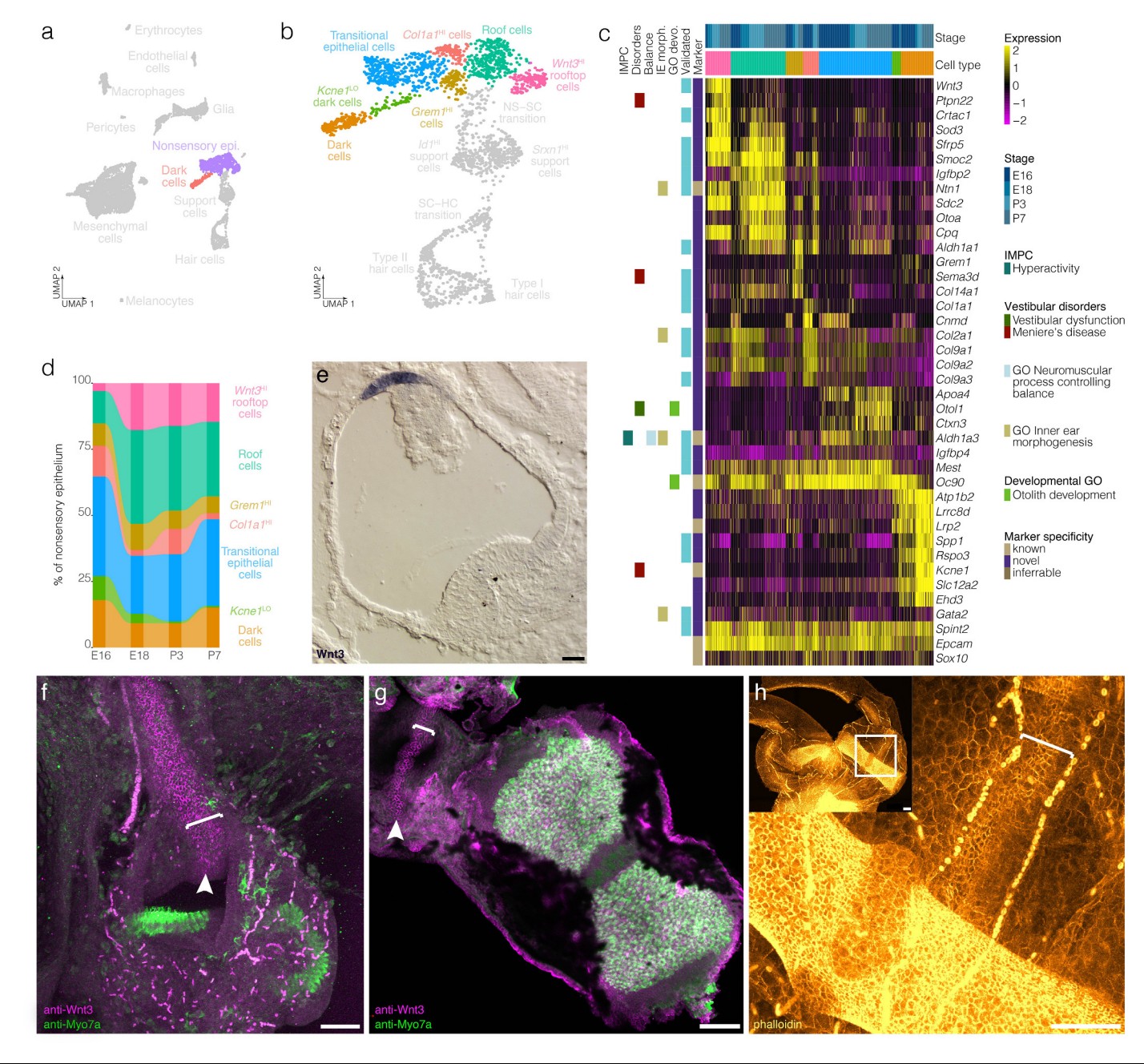

**Figure 4.** Cellular diversity in the nonsensory ampulla epithelium. (**a**) Highlights the position of the nonsensory epithelial cells of the ampulla in UMAP space relative to the whole dataset. (**b**) Cluster analysis in nonsensory ampulla epithelium. (**c**) Expression of markers relative to cell cluster and developmental time. Additional color bars indicate whether markers are known or novel in specificity or implicated in vestibular diseases and dysfunction. (**d**) The relative contributions of cell clusters to the nonsensory ampulla epithelium at the indicated developmental ages. (**e**) *Wnt3* RNA ISH localizes to a distinct population of cells in top of the roof of the P1 Swiss Webster ampulla. (**f**) Wnt3 IF in vibratome-sectioned crista localizes to the top of the roof of the P0 B6 ampulla and extends into the semicircular canal. (**g**) Wnt3 IF in a whole mount P7 B6 crista bisected to show the apical surface of the rooftop en face. (**h**) Phalloidin staining in P3 B6 ampulla whole mount. Note that roof cells in the Wnt3[+] domain are smaller than surrounding epithelial cells. *Brackets* in **f**–**g** indicate the Wnt3[+] rooftop domain. Scale bars = 50 μm.

The online version of this article includes the following figure supplement(s) for figure 4:

**Figure supplement 1.** Localization of cluster-specific markers in nonsensory ampulla epithelium.

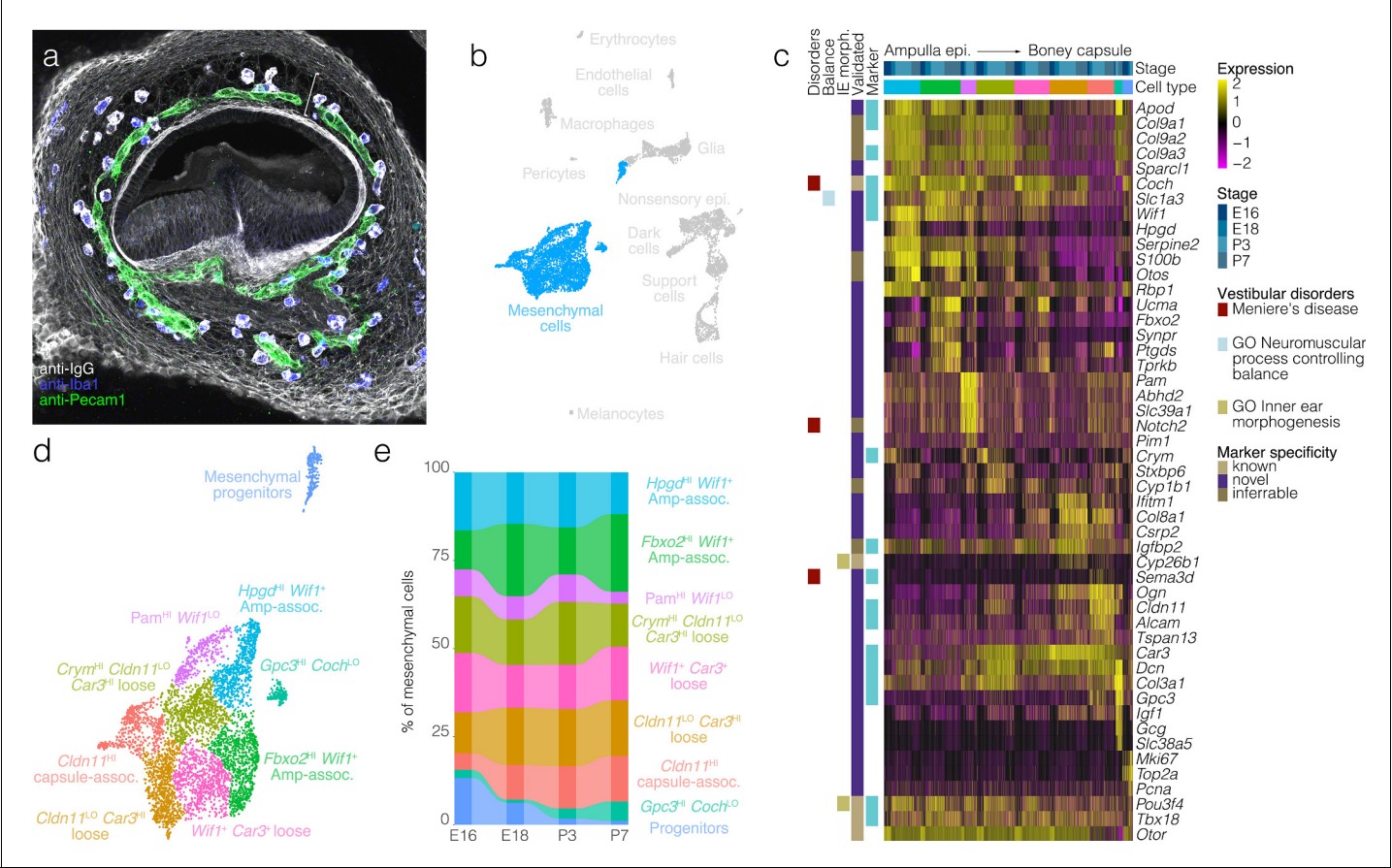

**Figure 5.** Mesenchymal cell diversity and dynamics. In (**a**), Labeling of endogenous IgG (*white*) in a vibratome section of E15.5 posterior crista demonstrates the region of loose mesenchyme (*bracket*) bounded by regions of dense mesenchyme proximal to the ampulla epithelium and the otic capsule cartilage. Iba1 (*blue*) and Pecam1 (*green*) IF demonstrate the macrophages and endothelial cells, respectively, present in the otic mesenchyme. (**b**) Highlights the position of mesenchymal cells in UMAP space relative to the whole dataset. Mesenchymal progenitors expressing both G2/M and mesenchymal markers group closely to proliferating glia in UMAP (see *Pou3f4* and *Mki67* expression in the UMAP in *Figure 1c*). (**c**) Expression of markers relative to cell cluster, developmental stage and localization in the ampulla relative to ampulla epithelium and boney capsule (see validation by RNA ISH in *Figure 5—figure supplement 1*). (**d**) Cluster analysis in mesenchymal cells. (**e**) The relative contributions of cell clusters to the composition of the mesenchyme at the indicated developmental stages.

The online version of this article includes the following figure supplement(s) for figure 5:

**Figure supplement 1.** Localization of cluster-specific markers in otic mesenchyme surrounding the ampulla.

**Figure supplement 2.** Slc1a3-CreER activity in the anterior canal, horizontal canal and utricle.

Analysis of cluster proportions shows a progressive loss of glial progenitors (*salmon pink*) and a corresponding increase in Schwann cells (*blue*) from E18 to P7 (*Figure 3e*, *Figure 3—figure supplement 1*). Allen Institute RNA ISH data independently validates the presence of *Npy*^HI cells in E15.5 B6 crista (*Figure 3—figure supplement 2*).

## Cellular diversity and developmental dynamics in the nonsensory ampulla epithelium and otic mesenchyme

Clustering detects transcriptionally distinct nonsensory epithelial cells in the crista ampullaris including transitional epithelial cells, dark cells and novel subtypes of the ampulla roof and wall (*Figure 4a–c*, see *Figure 4—figure supplement 1* for RNA-ISH validation). Transitional epithelial cells were identified based on expression of the known markers *Oc90* and *Ald1a3* (*Verpy et al., 1999*; *Ono et al., 2020*) and also on their expression of novel markers *Igfbp4* and *Mest* (*Figure 4b–c*, *Figure 4—figure supplement 1*). Dark cells were identified based on expression of the known markers *Kcne1* (*Warth and Barhanin, 2002*) and *Lrp2* (*Tauris et al., 2009*) and also on their

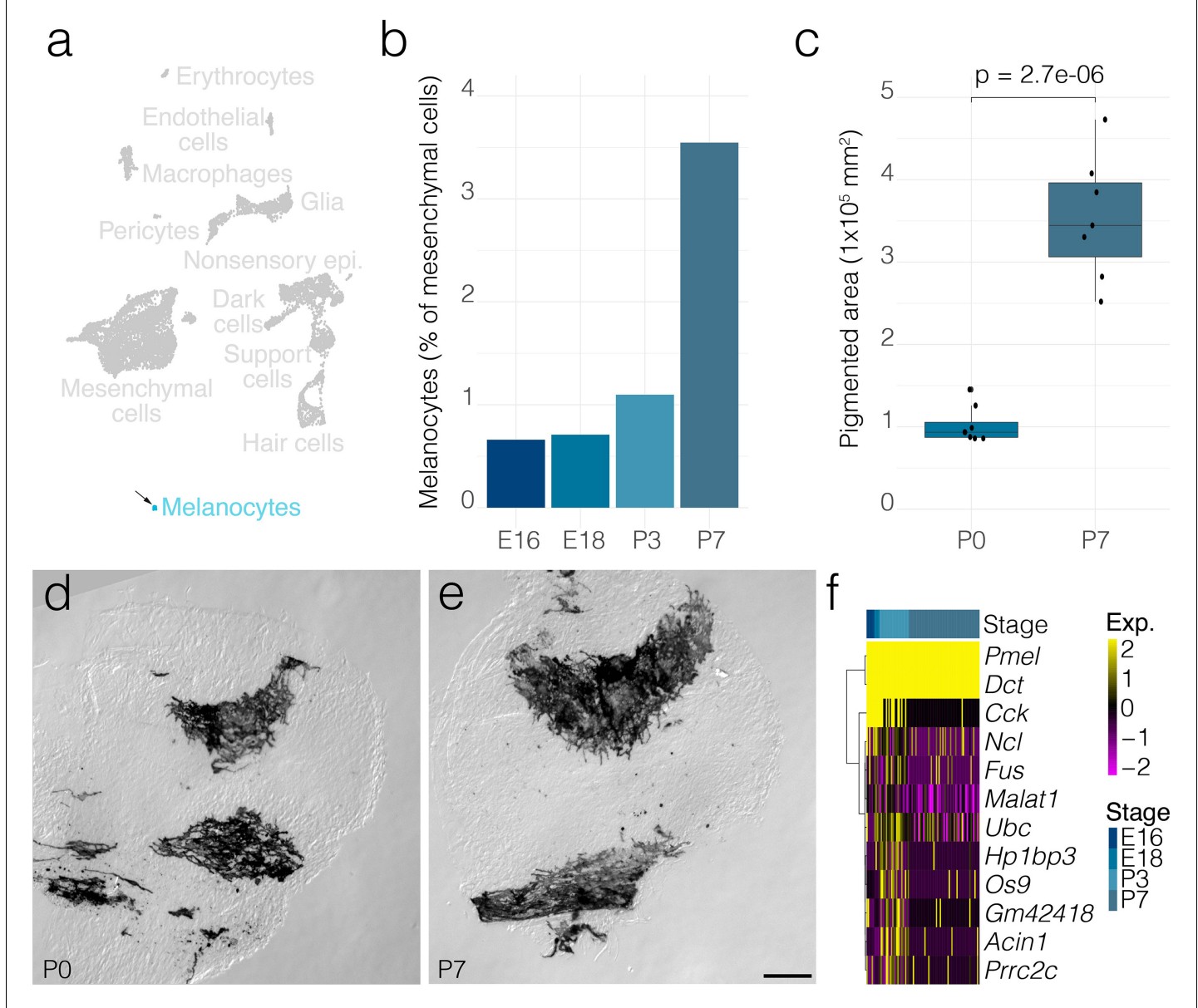

**Figure 6.** Melanocyte dynamics. (a) *Arrow* indicates the cluster of melanocytes in UMAP space relative to the whole dataset. (b) The relative contributions of melanocytes to the composition of the mesenchyme at the indicated developmental stages. (c) The area occupied by pigmented cells quantified from DIC micrographs of whole mount cristae at P0 (*n* = 7) and P7 (*n* = 7) as shown in d and e, respectively. (f) Shows differential gene expression in melanocytes at P7 relative to those at P3. Scale bar = 100 µm.

expression of novel markers including *Spp1* and *Rspo3* (*Figure 4b–c*, see *Figure 4—figure supplement 1* for RNA ISH validation). Trajectory and velocity analysis suggest that the cluster labeled as *Kcne1*[LO] dark cells represents a developmentally immature population of dark cells (*Figure 2e* and *Figure 2—figure supplement 3*). Analysis of cell cluster proportions in the nonsensory ampulla epithelial suggests increases in *Kcne1*[HI] dark cells relative to *Kcne1*[LO] dark cells between E16 and P7 (*Figure 4d*).

The remaining nonsensory epithelial cells comprise four clusters expressing markers also having spatially distinct expression patterns within the ampulla roof and wall (*Figure 4b–c*, *Figure 4—figure supplement 1*). The markers of the roof cell cluster identified by scRNA-seq *Smoc2*, *Igfbp2*, *Gata2*, and *Ntn1* localize to epithelial cells in the roof of the crista ampullaris (*Figure 4c*, *Figure 4—figure supplement 1*). By contrast, markers that are enriched in the UMAP-adjacent rooftop cell cluster—

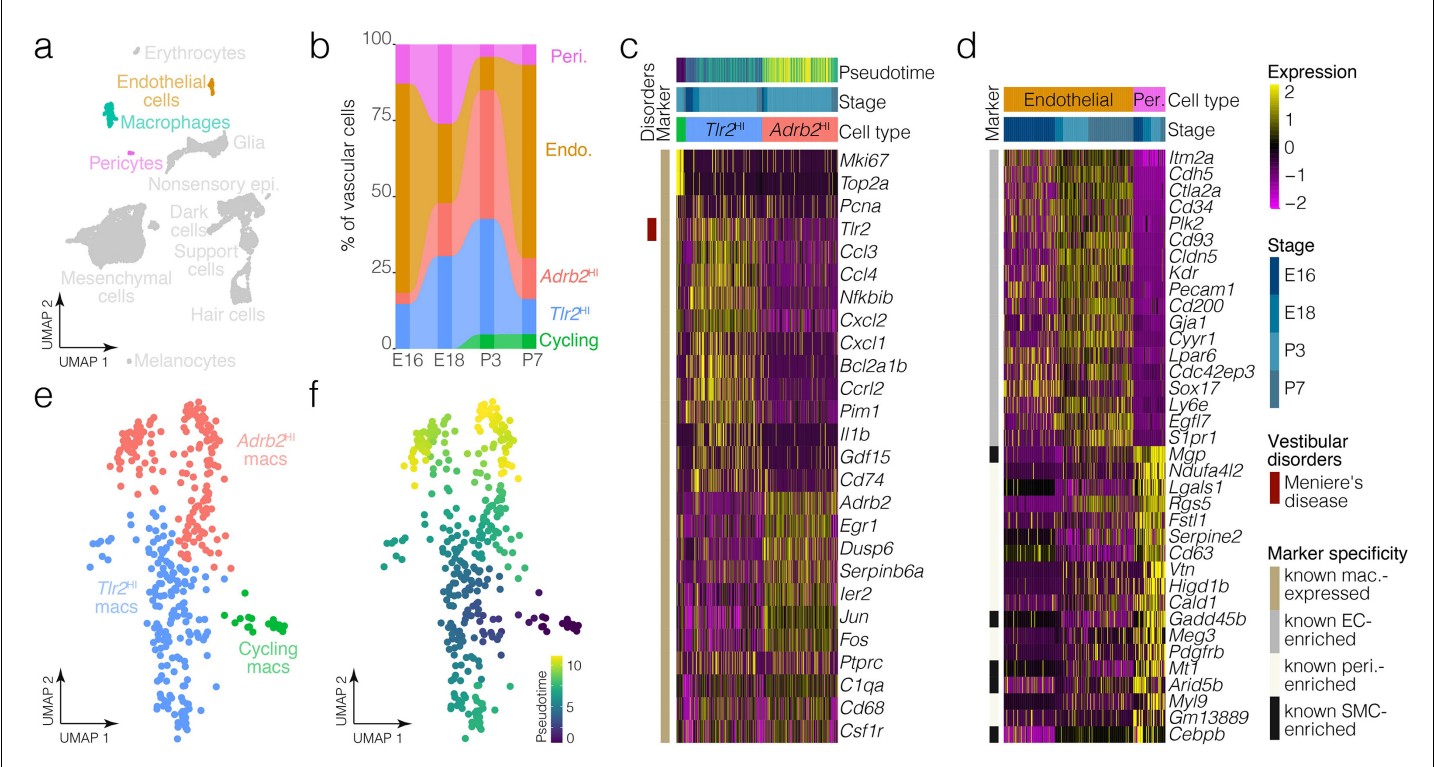

**Figure 7.** Macrophage and vascular cell diversity and dynamics. (**a**) Highlights the positions of macrophages and vascular cells in UMAP space relative to the whole dataset. (**b**) The relative contributions of cell clusters to the composition of the vasculature at the indicated developmental stages. A subset of macrophages is not associated with blood vessels (**Figure 5a**). (**c**) Expression of markers relative to macrophage cluster. (**d**) Relative expression of markers in endothelial cells and pericytes. (**e**) Cluster analysis in the macrophages. (**f**) Pseudotime of the macrophage trajectory. The online version of this article includes the following figure supplement(s) for figure 7:

**Figure supplement 1.** Localization of macrophage markers.

**Figure supplement 2.** Developmental stage relative to UMAP in ampulla macrophages.

*Wnt3*, *Crtac1* and *Sfrp5*—are restricted to the top of the roof of the ampulla (**Figure 4c,e–g**, **Figure 4—figure supplement 1**). *Wnt3* RNA ISH and IF localize to the rooftop (**Figure 4e–g**, **Figure 4—figure supplement 1**). Wnt3 IF extends throughout the inner rim of the semicircular canals in all three cristae (*not shown*). In addition to this distinct gene expression pattern, phalloidin staining shows that cortical actin rings in cells in the Wnt3 domain are smaller in diameter than surrounding roof epithelial cells (**Figure 4h**). Cluster-specific markers *Sema3d* and *Col1a1* localize to the walls of the ampulla (**Figure 4c**, **Figure 4—figure supplement 1**). Trajectory and velocity analysis suggest that the collagen-enriched wall cells may represent a developmentally immature population of the *Grem1*^HI *Sema3d*^HI wall cells (**Figure 2e** and **Figure 2—figure supplement 3**). Analysis of cell cluster proportions in the nonsensory ampulla epithelial suggests increases in *Grem1*^HI *Sema3d*^HI wall cells relative to collagen-enriched wall cells by P7

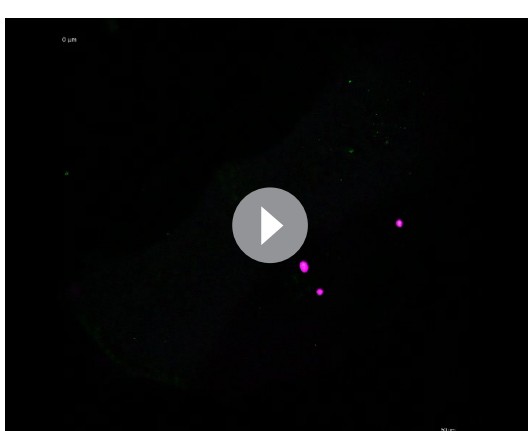

**Video 1.** Id1 immunofluorescence in P0 crista. Video shows a series of 1 micron optical sections per second imaged in P0 B6 cristae showing anti-Sox2 (*green*), anti-Id1 (*magenta*), and phalloidin (*cyan*).
https://elifesciences.org/articles/60108#video1

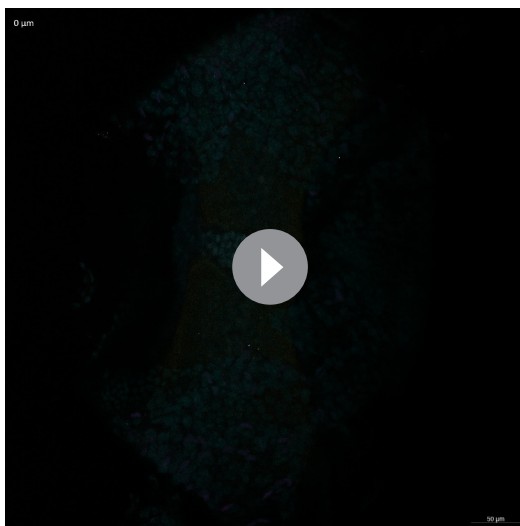

**Video 2.** Id1 immunofluorescence in P4 crista. Video shows a series of 1 micron optical sections per second imaged in P4 Hes5-GFP cristae from a colleagues' colony showing anti-Sox2 (white), anti-Id1 (*magenta*), Hes5-GFP (yellow), and Hoechst 33342 (*cyan*).
https://elifesciences.org/articles/60108#video2

(*Figure 4d*) and also increases in roof and roof-top cells.

Mesenchyme surrounding the crista ampullaris consists of histologically distinct spatial domains: dense mesenchyme associated with the ampulla epithelium, loose mesenchyme containing the vasculature and dense mesenchyme near the boney otic capsule (*Figure 5a*). Clustering detects transcriptionally distinct subtypes of mesenchymal cells corresponding to these three spatial domains (*Figure 5c*, see *Figure 5—figure supplement 1* for RNA-ISH validation). For example, the dense mesenchyme associated with the ampulla epithelium expresses *Wif1*, *Coch*, *Col2a1*, *Col3a1*, *Col9a1*, *Col9a3*, *Igfbp2*, and *Slc1a3* but lacks *Car3, Alcam, Sema3d*, and *Dcn*. By contrast, loose mesenchyme expresses *Car3*, *Dcn*, *Coch*, and *Igfbp2* and lacks expression of *Wif1*, *Col2a1*, *Col9a1*, *Col9a3*, *Slc1a3*. The dense mesenchyme near the boney otic capsule expresses *Cldn11*, *Gpc3*, *Sema3d*, *Car3*, *Dcn*, and *Col3a1* and lacks expression of *Wif1*, *Col2a1*, *Col9a1*, *Col9a3*, and *Slc1a3*. Like *Slc1a3* expression in single-cell RNA-seq and RNA ISH, Slc1a3-CreER activity extends to mesenchymal cells (*Figure 5—figure supplement 2*). Additional clusters identified include those labeled as mesenchymal

progenitors based on enrichment with cell cycle markers such as *Mki67* (*Figure 5c*, see *Figure 1c* for *Mki67* in UMAP) and a cluster of loose mesenchyme enriched in *Crym*, which localizes only on the neural side of the ampulla mesenchyme in Allen RNA ISH images (*Figure 5c*, *Figure 5—figure supplement 1*). Proportions of mesenchymal progenitors decrease incrementally from E16 to P7 (*Figure 5e*).

In addition to fibroblast-like cells, ampulla mesenchyme includes melanocytes (*Figure 6*). As a proportion of total mesenchymal cell numbers, melanocytes increase ~3.5-fold by P7 from E18 (*Figure 6b*). Similarly, the area occupied by pigmented cells increases ~3.5-fold in P7 versus P0 crista ampullaris (*Figure 6c–e*). Several genes showed downregulation in P7 melanocytes versus P3 (*Figure 6f*).

Ampulla mesenchyme also includes blood vessels and macrophages; many macrophages in the ampulla localize to blood vessels (*Figure 5a*). Clustering detects endothelial cells, pericytes and distinct subtypes of macrophages in the crista ampullaris (*Figure 1*, *Figure 7*). $Tlr2^{HI}$ and $Adrb2^{HI}$ macrophage clusters are both present at all stages examined (*Figure 7c*), whereas macrophages enriched in *Mki67* and other cell cycle markers were only present at P3 and P7. $Tlr2^{HI}$ and $Adrb2^{HI}$ macrophage subtypes may represent inflammatory states. For example, $Tlr2^{HI}$ macrophages express *Tlr2, Ccl3, Il1b* and other cytokine-signaling ligands whereas $Adrb2^{HI}$ macrophages are enriched in immediate-early targets including *Egr1*, *Ier2*, *Jun*, and *Fos*. RNA ISH images from the Allen Inst. independently validate the presence of a sparse population of $Egr1^+$ cells in the loose mesenchyme surrounding the E15.5 ampulla (*Figure 7—figure supplement 1*). Expression patterns in endothelial cells versus pericytes (*Figure 7d*) are consistent with those in the Human Protein Atlas (*Uhlén et al., 2015*) and in scRNA-seq data from brain (*He et al., 2018*). Analysis of the proportions of macrophage and vascular cell clusters suggests macrophage dynamics in the E16-P7 window (*Figure 7b*, *Figure 7—figure supplement 1*).

## Comparative analysis of cochlear and crista hair cell and support cell expression

The hair and support cells of the cochlea and the crista have similarities but are specialized functionally, which suggests that similarities and differences can be found at the transcriptional level. To

compare the gene expression profile of crista hair cells and support cells to those in the cochlea, we aligned a publicly available dataset of P7 cochlea cells from *Kolla et al., 2020* to the P7 crista cells of the present study, then clustered the cells, identified cell type clusters based on known markers and performed differential expression analysis. UMAP and clustering suggest that crista cells including support cells and hair cells resemble cochlear cells (*Figure 8a*). Heatmaps show shared and specific markers for crista vs. cochlea support cells and hair cells identified by differential expression analysis (*Figure 8b*). Differential expression analysis identifies known cochlea-specific support cell markers *Tecta* and *Tectb*, the known crista-specific support cell marker *Zpld1*, additional markers identified in previous studies comparing cochlear hair cells and support cells to vestibular hair cells and support cells (*Elkan-Miller et al., 2011*; *Scheffer et al., 2015*; *Giffen et al., 2019*), as well as several novel crista-specific markers such as *Agr3*, *Meis2* and *Bricd5* in support cells and some immature hair cells and *Pcdh20* and *Cib3* in hair cells.

## Expression of genes associated with vestibular dysfunction and disease in the crista ampullaris

For insights into the cellular players in vestibulopathies, in *Figure 9—source data 1* we show cluster analysis of expression in crista ampullaris cells of genes associated with vestibular diseases and disorders (i.e. Meniere's disease, vertigo, motion sickness, circling, and hyperactivity in mice and relevant gene ontology terms). Clustering shows that many of the genes associated with vestibular disorders are expressed predominantly in support cells and hair cells (*Figure 9* shows a subset, *Figure 9—source data 1* shows all vestibular disorder genes). However, some of these genes implicated in

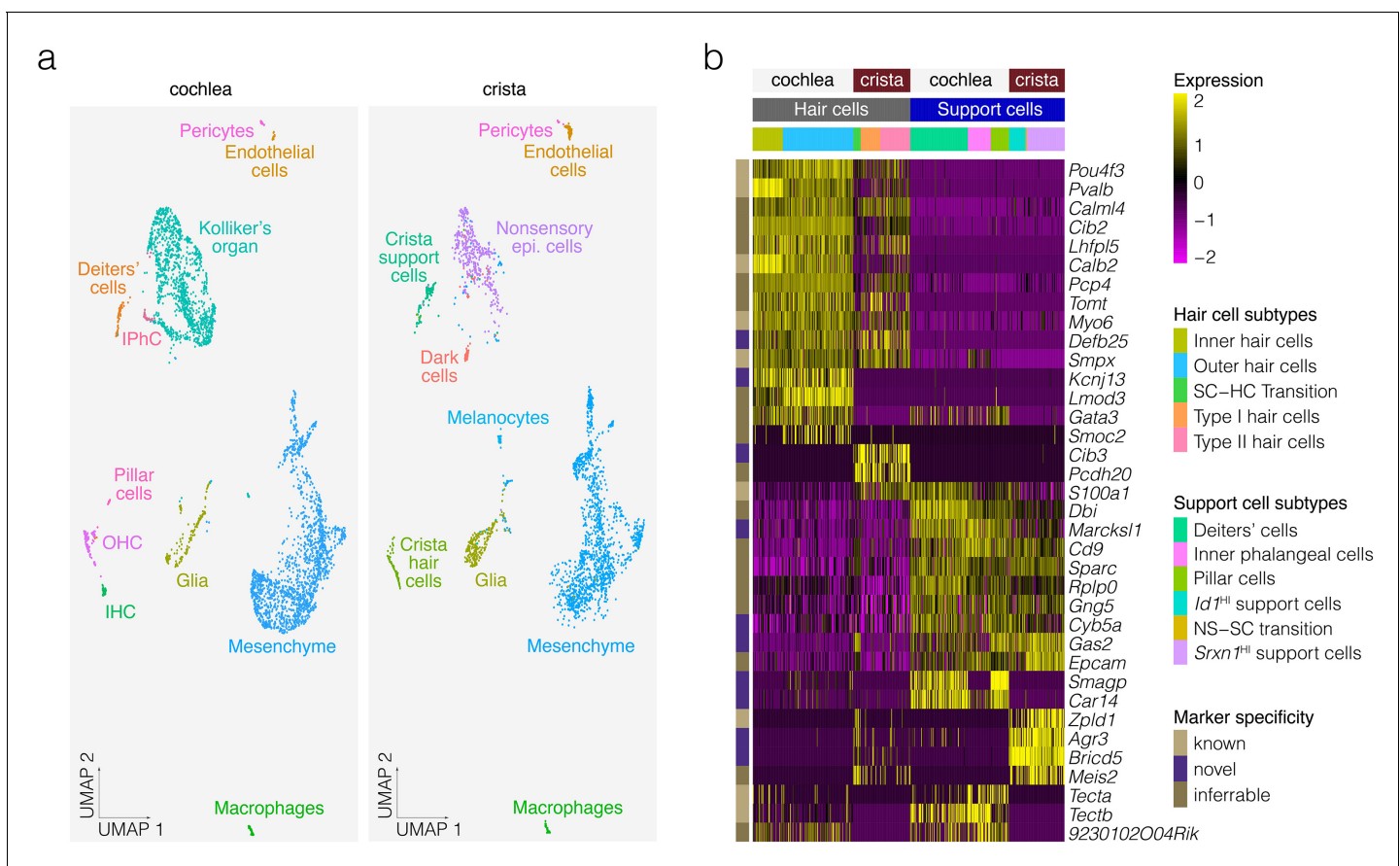

**Figure 8.** Comparative analysis of P7 crista ampullaris and P7 cochlea. (**a**) Shows the relatedness of scRNA-seq data from Kolla et al. for P7 Swiss Webster cochlea (*left*) and the P7 B6 crista ampullaris data from the present study (*right*) in UMAP space. Note that common cell types overlap, indicating similarity, in contrast to tissue-specific cell types such as dark cells and pillar cells. (**b**) Shows differential expression in the hair cells and support cells of the cochlea vs. those of the crista.

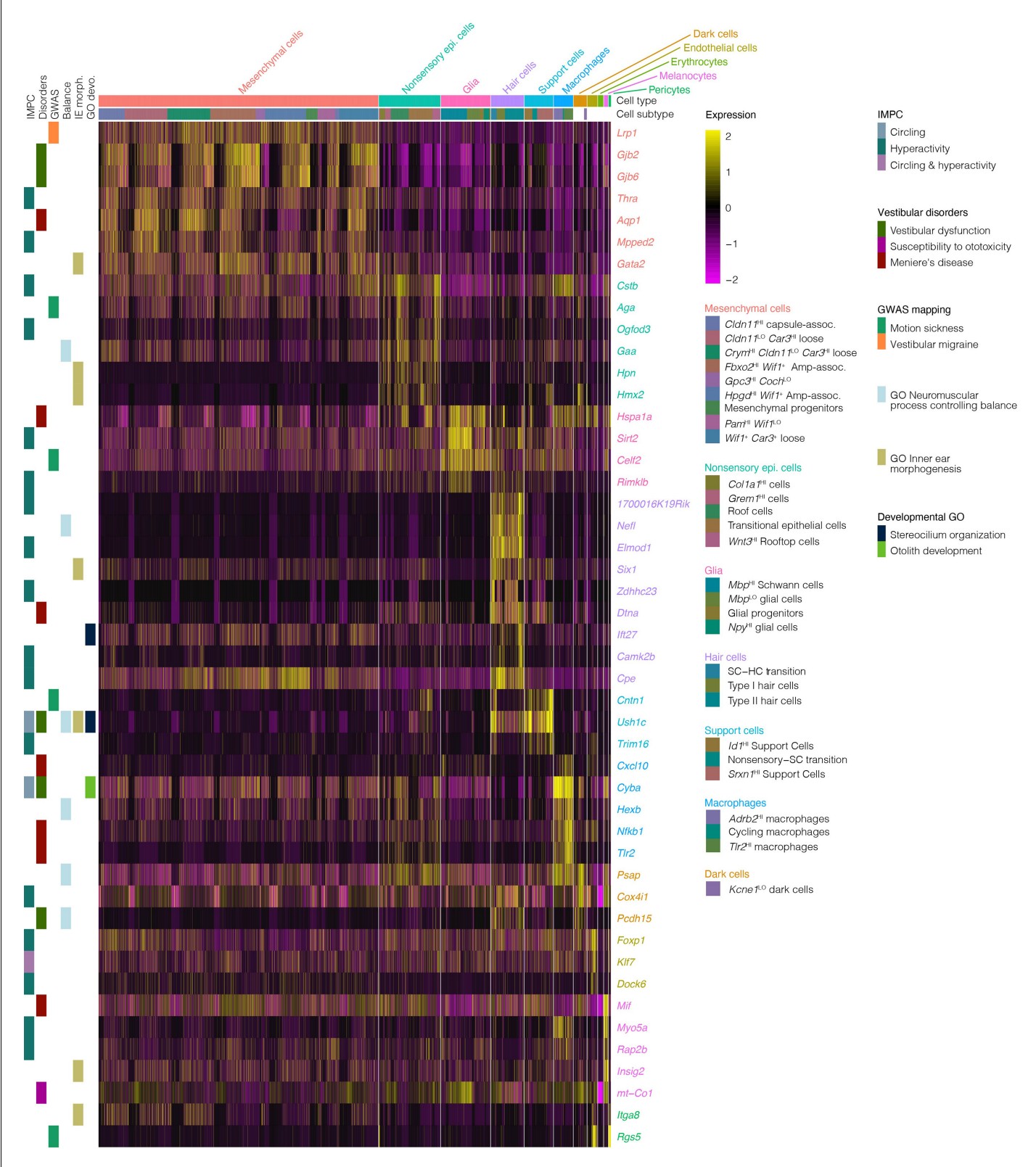

**Figure 9.** Expression of genes associated with vestibular disease and dysfunction in the crista ampullaris. The heatmap summarizes the expression of a subset of genes associated with vestibular disease and dysfunction in cell clusters of the crista ampullaris. *Figure 9—source data 1* shows the full list of vestibular disorder-associated genes. Column color bars indicate the major cell types and subtypes. Row color bars denote the following: *IMPC*, genes mutated in mice that exhibit circling (*cornflour blue*) and hyperactivity (*teal*) or both (*purple*) phenotypes. *Disorders*, gene associations with Vestibular

*Figure 9 continued on next page*

*Figure 9 continued*

dysfunction (*forest green*), susceptibility to ototoxicity (*magenta*) and Meniere's disease (*maroon*) were compiled from MSigDB, OMIM and Malacards databases. *GWAS*, hits in genome-wide association studies of motion sickness (*green*) and vestibular migraine (*orange*). *Balance*, genes in the GO term 'neuromuscular process controlling balance' (*sky blue*). *IE morph*, genes in the GO term 'inner ear morphogenesis' (*khaki*). *GO devo.*, genes in GO terms 'stereocilium organization' (*midnight blue*) and 'otolith development' (*lime green*).

The online version of this article includes the following source data for figure 9:

**Source data 1.** Expression of genes associated with vestibular disease and dysfunction in the crista ampullaris.

vestibular dysfunction are also expressed in other cell types of the ampulla. For example, macrophages show enrichment for *Cstb*, *Cxcl10*, *Atg5*, *Cyba*, *Hexb*, *Nfkb1*, *Tlr2*, *Pepd*, and *Nrp1*. Glia show enrichment for *Hspa1a*, *Sirt2*, *Celf2*, and *Rimklb*. Melanocytes show enrichment for *Mif*, *Myo5a*, *Rap2b*, *Insig2*, and *mt-Co1*. Endothelial cells show enrichment for *Foxp1*, *Klf7*, *Rgs5*, *Nrp2*, *Zeb1*, and *Dock6*. Mesenchymal cells show enrichment for *Lrp1*, *Gjb6*, *Wnt5a*, *Slc1a3*, *Aqp1*, *Zic1*, *Fgfr4* and more.

## Discussion

scRNA-seq is a powerful technology for studying cellular diversity and changes in developing tissues. Our results provide several new insights into the composition and development of the crista and identify markers for novel subtypes of support cells, nonsensory epithelial cells, macrophages, glia, and mesenchymal cells. Most notably, we found cell cluster-specific markers that predicted stereotypic gene expression domains not previously described in the ampulla epithelium and mesenchyme. Second, this analysis resolves novel developmental transitions at the transcriptional level including the separate lineages of type I and type II hair cells, the maturation of Scwann cells as well as evidence for nonsensory–support cell conversion and macrophage diversity and local proliferation. Third, we showed this dataset can be leveraged to identify the stage- and lineage-dependent transcriptional changes during the formation of hair cells in the crista. We have uploaded this dataset to GEO (GSE168901) for independent investigation and reanalysis as the informatics tools and research questions continue to evolve.

### Support cell subtypes and postnatal hair cell addition in the crista

Several findings increase our understanding of hair cell recruitment from precursors in the support cell layer of the developing crista. Findings from trajectory analysis, the changes in cluster proportions and RNA velocity analysis (*Figure 2* and *Figure 2—figure supplement 3*) suggest recruitment of hair cells from the support cell layer. Transitioning cells are Atoh1$^+$ and we found these cells in the support cell layer by IF in neonatal cristae. Based on the expression of both support cell and hair cell markers and the basal position of their Atoh1$^+$ nuclei, transitioning cells likely represent the immature hair cells previously identified in utricle based on morphological characteristics including nuclei in the basal layer (*Rüsch et al., 1998*; *Li and Forge, 1997*).

Our findings increase the understanding of hair cell differentiation in the vestibular system. Whereas most Anxa4$^+$ type II hair cells form postnatally in the utricle (*McInturff et al., 2018*; *Burns et al., 2012*), we identify an Anxa4-enriched cluster of hair cells in E16 scRNA-seq dataset as well as in the hair cell layer of E15.5 RNA ISH images of crista (*Figure 2*, *Figure 2—figure supplement 2*). Our analysis provides evidence that type I and type II hair cells differentiate from a common immature hair cell population and we have identified differential gene expression in the type I and type II lineage including *Sox21* and *Insm1*, respectively. Insm1 is a transcriptional repressor required for outer hair cell differentiation (*Wiwatpanit et al., 2018*) but its function in the vestibular system has not been examined to our knowledge. Our finding raises the question of whether Insm1 suppresses type I hair cell differentiation in developing type II hair cells of the crista similar to its role in the cochlea, where it represses inner hair cell-enriched genes in nascent outer hair cells.

Between E18 and P7, clusters of type I hair cells, type II hair cells and *Srxn1*$^{HI}$ support cells increase as a percentage of sensory epithelial cells while *Id1*$^{HI}$ support cells and transitional cells decrease. One possible explanation for the hair cell increase is an increase in hair cell differentiation during this period. We examined the first possibility by immunolabeling and manually counting hair cell numbers to confirm that hair cells indeed increase postnatally. These findings are consistent with

postnatal decline in immature hair cells numbers reported in the utricle (*Rüsch et al., 1998*). It follows that support cells should decrease. scRNA-seq showed decline in *Id1*HI support cells, which we confirmed independently by manual counts of Id1+ nuclei in situ. An alternative explanation for these changes is that the *Id1*HI support cells and *Srxn1*HI support cells represent immature and mature support cells but RNA velocity does not indicate such a transition. These findings support that some support cells remain competent to undergo differentiation into hair cells and thus represent a potential cellular target for promoting hair cell regeneration through cellular reprogramming. In mammals, regeneration in the vestibular organs is slow, highly variable and limited to Type II-like hair cells with immature hair bundle morphology (*Kinoshita et al., 2019*; *Sayyid et al., 2019*; *Forge et al., 1993*). Mechanistically, support cell–hair cell conversion may relate to transient gene expression in the transitional cells. The transitional cell-enriched genes *Isl1*, *Sox2*, and *Sox21* are known to regulate support cell–hair cell conversion in the cochlea (*Yamashita et al., 2018*; *Hosoya et al., 2011*; *Dabdoub et al., 2008*). Additionally, Notch targets *Hes1*, *Hey1*, *Jag1*, *Igfbp3*, and *Slitrk6* (*Campbell et al., 2016*) were elevated in early postnatal support cells and transitional cells, then declined in maturing hair cells. The transient expression of Notch is consistent with the contrasting positive early role and later negative role of Notch-signaling in hair cell differentiation (*Campbell et al., 2016*; *Hartman et al., 2010*), the perinatal window of greater regeneration-competence in support cells (*Slowik and Bermingham-McDonogh, 2013a*) and earlier results demonstrating competence in a subset of support cells to convert into hair cells in response to Notch inhibition (*Slowik and Bermingham-McDonogh, 2013b*).

We have identified specific expression patterns of Fgf ligands in the crista and found that Fgf7 and Fgf21 show specificity to the two types of hair cells. Furthermore, we find Fgf7 in the transition to type I hair cells and Fgf21 in transition to type II cells. The roles of Fgf7 and Fgf21 in inner ear development and function are unknown as far as we know. Findings from other systems implicate Fgf7 in presynaptic organization. For example, Fgf7 promotes clustering of anti-synapsin+ vesicles and neurite branching in cultured motorneurons (*Umemori et al., 2004*). In vivo, *Fgf7* deletion in mice is associated with normal motor function but altered nociceptive responses in sensory neurons of the dorsal root ganglion (*Liu et al., 2015*). Furthermore, *Fgf7* knockout mice have increased susceptibility to epilepsy associated with increased hippocampal neurogenesis and mossy fiber sprouting and decreased inhibitory vesicles in pyramidal neurons (*Terauchi et al., 2010*; *Lee et al., 2012*). In contrast to Fgf7, Fgf21-signaling through its receptor Fgr1 and coreceptor Klb promotes insulin sensitivity and glucose uptake (*Kliewer and Mangelsdorf, 2019*), which are broadly important for neurodevelopment and function. Given *Fgf7* expression in crista type I hair cells and the evidence for its specific role in neural function, we are testing whether signaling of Fgf7 through its receptor Fgfr2 influences the vestibular afferents. Currently, we are also utilizing this dataset to identify candidate regulatory genes and pathways for cellular reprogramming to promote hair cell regeneration.

To our surprise, inducible Hsp70 IF is present in apical processes extending from a subset of support cells at P7 but not at P3. The anti-Hsp70 foci are adjacent both to hair bundles and the apex of a subset of support cells. It is unclear from which cells these foci originate. When we initially found upregulation of heat-shock response genes in support cells by P7, we hypothesized that heat-shock response was likely artifactual and caused by stress from dissociation. However, when we acutely dissect cristae, fix and stain, we find that Hsp70 indeed increases, suggesting that Hsp70 is upregulated by P7 in the crista. Upregulation of expression of heat shock response genes was observed at P7 in hair cells, support cells, nonsensory epithelial cells by scRNA-seq. Exosomal secretion of Hsp70 from support cells promotes hair cell survival in the utricle and cochlea (*Breglio et al., 2020*; *May et al., 2013*; *Taleb et al., 2009*). Additionally, there is an association of a single-nucleotide polymorphism in *HSPA1A/HSP70-1* with Meniere's disease (*Kawaguchi et al., 2008*). The role of Hsp70 in inner ear development is an open question that could be addressed in future studies.

## Novel cell types in the ampulla roof and possible functions

scRNA-seq analysis resolved detailed expression profiles for transitional epithelial cells and dark cells and for novel cell types in the roof of the nonsensory ampulla epithelium. Further study will be needed to determine the precise relationships between the cell-specific gene expression pattern identified by cluster analysis and the morphological variations (e.g., microvilli, granules, cell shape, and osmiophilicity) identified by earlier histological analysis in ampulla epithelium. Regardless, these results provide insights into the development and function of the ampulla epithelium. Nonsensory

ampulla epithelial cells and support cells express the peptidase *Spint2*, which inhibits the proteolytic activation of Hgf (*Kawaguchi et al., 1997*). Hgf-signaling is required for melanocyte invasion of the nonsensory cochlear epithelium (*Shibata et al., 2016*). By extension, Spint2 may act to inhibit melanocyte invasion of the ampulla epithelium through inhibition of Hgf/c-met-signaling in melanocytes.

The cellular heterogeneity found in the ampulla epithelium may reflect differences in developmental lineage. Cristae derive from the dorsal otocyst (*Li et al., 1978*), with sensory specification evident by E11.5–12 (*Morsli et al., 1998*). At E12-E13, outpockets of the dorsal otocyst termed the canal pouches remodel to form the tube-like epithelium of semicircular canals and ampullae. This process involves apical fusions of the central walls of the pouches to form bilayered epithelium termed the canal fusion plates, then basal detachment and resorptions of the epithelial cells in the canal fusion plates into the rims of the pouches to form the canals (*Morsli et al., 1998*; *Martin and Swanson, 1993*; *Rakowiecki and Epstein, 2013*; *Salminen et al., 2000*). These cellular movements that remodel the canal fusion plate into the semicircular canals are regulated through morphogenetic gradients in signaling of Wnt, Notch, Bmp and Netrin 1 (Ntn1) within the rims and plates (*Kiernan et al., 2001*; *Rakowiecki and Epstein, 2013*; *Salminen et al., 2000*; *Chang et al., 2008*). For example, *Ntn1* expression in the canal fusion plates is required for basal detachment and resorption (*Salminen et al., 2000*). Expression of *Ntn1* persists in the inner rim of the canals and in the roof of the ampulla. Thus, transcriptional heterogeneity within the canal and ampulla may relate to regional restrictions in signaling and expression in the canal rims and fusion plates. Single cell transcriptomic analysis of earlier embryonic cristae would likely resolve these important developmental transitions. scRNA-seq analysis also identified a *Wnt3*-specific expression in a cluster of cells in the roof of the ampulla. Wnt-signaling is present throughout the dorsal otocyst by E11.5 (*Rakowiecki and Epstein, 2013*; *Noda et al., 2012*), then persists in the vestibular epithelium and increases in response to injury (*Wang et al., 2015*). Whereas *Wnt1* and *Wnt3a* expression in the hindbrain was demonstrated to be required for induction of otocyst development, other functional redundancies have hampered the identification of critical sources of Wnt for later Wnt-dependent processes (*Jansson et al., 2015*) such as orientation of hair cell stereocilia bundles (*Hua et al., 2014*), the formation of the canal fusion plate (*Rakowiecki and Epstein, 2013*; *Noda et al., 2012*) and otic mesenchyme differentiation (*Bohnenpoll et al., 2014*). Thus, the Wnt3[+] rooftop cells may act as a morphogen source for Frizzled-signaling. Regarding canal formation, *Wnt3* is expressed coincidently with *Nr4a3* and *Ntn1* in the dorso-lateral E11.5 otocyst (*Appendix 1—figure 1*). Nr4a3/ Nor1 (*Ponnio et al., 2002*) and Ntn1 (*Salminen et al., 2000*) activity in the canal plate/canal inner rim may relate to Wnt3-signaling. Also of relevance to the question of regional Wnt activity in the crista is that Wnt-signaling regulators showed specificity for cell types in the ampulla (e.g. see in *Appendix 1—figure 2* the Wnt inhibitor *Wif1* in ampulla-proximal dense mesenchyme, *Wnt7b* in nonsensory epithelium, the Wnt activator *Rspo3* in dark cells). The role of rooftop cells in cupula function and in vestibular disorders is unknown.

## Schwann cell development and implications for the study of demyelinating disease

Glia in the cochleovestibular nerve are derived from the neural crest (*Sandell et al., 2014*) and otic placode (*Xu et al., 2017*). In mouse, glia are known to proliferate within the cochleovestibular nerve up to birth (*Ruben, 1967*) and in response to injury (*Lang et al., 2011*). Consistent with this, many glia express G2/M markers at E18 and these glial progenitors decline by P7. Furthermore, myelination markers such as *Mbp* and *Mpz*, *Egr2* (*Bosio et al., 1996*; *Jessen and Mirsky, 2008*) increase along the trajectory extending from glial progenitors. The transcriptional dynamics in Schwann cell differentiation resolved by scRNA-seq include genes associated with congenital demyelinating diseases such as *Pmp22* (Charcot Marie Tooth and Deafness Syndrome) and *Plp1* (Pelizaeus-Merzbacher disease). Future studies could leverage the transcriptomic profile of Schwann cell development for insights into pathological processes such as Schwann cell dedifferentiation after injury and Schwann cell transformation in vestibular schwannoma.

## Implications of mesenchymal cell diversity in the ampulla

Our analysis resolved detailed expression profiles for several previously unknown subtypes of mesenchymal cells. Specifically, we identified three types of mesenchyme: dense mesenchyme associated

with the ampulla epithelium, dense mesenchyme near the boney otic capsule and, in between, loose mesenchyme containing vasculature. Cluster specificity correlated corresponded well to spatial specificity in ISH (Allen Atlas) of cluster-specific markers revealing stereotypic patterns of localization of the subtypes both along a spatial axis between ampulla epithelium and the cartilaginous otic capsule and in loose ($Car3^{HI}$) versus dense ($Car3^{LO}$) mesenchyme. The identification of spatially distinct mesenchymal cell subtypes has potential implications for understanding how mesenchyme supports crista development and function. Whether mesenchymal cell subtypes of the ampulla are altered in vestibular disease as the mesenchymal cell subtypes in the cochlea are in hearing loss (*Hequembourg and Liberman, 2001*) is another open question.

Macrophages show enrichment for several genes associated with vestibular disorders including Meniere's disease, which raises the question of whether these cells have important roles in vestibular disease. Furthermore, cluster analysis resolved two transcriptionally distinct subtypes of macrophages: $Tlr2^{HI}$/$Ccl3^{HI}$/$Ccl4^{HI}$/$Nfkbib^{HI}$/$Cxcl2^{HI}$ macrophages and $Adrb2^{HI}$/$Dusp6^{HI}$/$Egr1^{HI}$/$Folr2^{HI}$ macrophages. Enrichment of immediately-early target genes was recently reported to be an experimental artifact in studies of microglia (*Li et al., 2019*), however, RNA ISH images from the Allen Institute corroborate the presence of a sparse Egr1$^+$ cell population in the ampulla mesenchyme. Whether the expression signatures of ampulla macrophage subtypes relate to localization, morphology, inflammatory activation, and developmental origins will be interesting questions to explore. We also found macrophages enriched for cell cycle markers, raising the possibility that some macrophages form locally within the developing ampulla, in contrast those recruited after hair cell injury (*Kaur et al., 2015*).

In our dataset, we did not have sufficient numbers endothelial cells and pericytes to make fine distinctions. Clustering can make tenuous divisions in small groups of cells but these 'over-clustered' groups had no convincing specific markers. Enrichment for these cells and other rare cells prior to scRNA-seq may resolve novel subtypes as has been shown for endothelial cells and pericytes in other systems (*He et al., 2018*).

## Conclusions

Our findings add to the fundamental understanding of the cellular diversity of the ampulla. Several novel subtypes of epithelial and mesenchymal cells were identified using scRNA-seq: Id1$^+$ and $Srxn1^{HI}$ support cells, support cells actively transitioning into hair cells, wall, roof and Wnt3$^+$ rooftop cells in the nonsensory ampulla epithelium, a spectrum of mesenchymal cell subtypes showing stereotypic localization within the otic capsule, $Tlr2^{HI}$ and $Adrb2^{HI}$ macrophages and progenitors of both glial cells and mesenchymal cells. The roles of these new cells in vestibular development, function and disease have yet to be elucidated. The specific markers of cell subtypes will now allow the study of cellular dynamics in ampulla development and vestibular disease.

## Materials and methods

### Mice

For scRNA-seq, timed pregnant C57BL/6J mice (B6; Jackson stock: 000664, RRID:IMSR_JAX: 000664) were purchased and aged to E18, P3, and P7. For the E16 sample, Sox2-GFP mice (Jackson stock: 07592) were bred to generate a timed pregnant litter. For most immunofluorescence (IF) studies, C57BL/6J mice were bred to generate timed pregnant litters. Two supplementary images show tissue collected opportunistically from Slc1a3-CreER:LNL-tTA:tetO-mAscl1-ires-GFP mice (*Todd et al., 2020*) and Hes5-GFP (*Nelson et al., 2011*) lines used in our colleagues' published research. To induce CreER, tamoxifen (1.5 mg in 100 μL of corn oil) was administered to adult mice daily for 5 days. Stages were verified by Theiler's criteria. Mice were housed in the University of Washington Department of Comparative Medicine. All procedures were reviewed and approved by the Institutional Animal Care and Use Committee of the University of Washington and performed in accordance with NIH guidelines.

### Dissociation of cells

All three cristae, including the ampullae, were dissected from E16 mice (n = 6), E18 mice (n = 5), P3 mice (n = 8), and P7 mice (n = 12) in ice cold Hank's buffered salt solution (HBSS; Cat. No. 14025–

92; Thermo Fisher Scientific; Waltham, MA). For the E18, P3, and P7 samples, half of the cristae were dissected away from the ampulla to increase the proportion of sensory cells. Scarpa's ganglion was not included. Dissected cristae and cristae with ampulla intact were then pooled (1:1) and dissociated by treatments with collagenase and papain as follows. First, cristae were digested in 2% collagenase IV (Worthington Biochemical; Lakewood, NJ) at 37°C for 5 min. A 10% vol of FBS was added and cristae were washed in HBSS three times. Second, cristae were digested in 2% collagenase II (Worthington) at 37°C for 30 min. As before, a 10% vol of FBS was added and cristae were washed in HBSS three times. Third, cristae were dissociated to single cells using the papain dissociation kit (Worthington, #LK003150) for approximately 1 hr with trituration every 10 min. then stopped with the addition of ovomucoid and spun per the manufacturer's instructions. Cells were strained through the cell strainer and counted with a hemocytometer. A total of 7000 cells were then input into the 10X protocol.

## scRNA-seq and analysis

Libraries for scRNA-seq were prepared using the Chromium Single Cell 3' Library and Gel Bead Kit v3 (10x Genomics; Pleasanton, CA) per the manufacturer's instructions. Reads were aligned to mm10 and filtered using the Cell Ranger pipeline. Cell numbers and depth are reported in *Supplementary file 1*. We used Monocle 3 (*Cao et al., 2019*), Seurat 3 (*Stuart et al., 2019*), velocyto (*La Manno et al., 2018*), and scVelo (*Bergen et al., 2020*) packages for scRNA-seq analysis. Normalization (SCTransform *Hafemeister and Satija, 2019*), batch correction (batchelor *Haghverdi et al., 2018*/IntegrateData), principal component analysis and dimensional reduction (UMAP *McInnes et al., 2018*) were performed in Seurat. Clustering (Leiden *Traag et al., 2019*) and trajectory analysis (learn_graph) were performed in Monocle 3. RNA velocity was analyzed using velocyto and scVelo. For heatmaps and FeaturePlots, raw counts were corrected for depth of sequencing and batch effects and scaled and centered using ScaleData in Seurat 3.1. Gene expression was plotted using pheatmap v1.0.12. Differential expression analysis was performed on raw counts using FindMarkers within Seurat 3.1. Gene set enrichment analysis (GSEA *Subramanian et al., 2005*) was performed on fold differences in expression using the fgsea package (*Sergushichev, 2016*) as described previously (*Wilkerson et al., 2019*) for canonical pathways (i.e. KEGG *Kanehisa et al., 2019*, Reactome *Jassal et al., 2020*, Biocarta *Nishimura, 2001* and PID *Schaefer et al., 2009*) and transcription factor targets (*Xie et al., 2005*) from MSigDB v7.0 (*Liberzon et al., 2015*). Gene lists for vestibular disease and dysfunction were curated by supplementing gene lists from Malacards (*Rappaport et al., 2017*), OMIM (*Amberger et al., 2019*), MSigDB v6.2 (*Liberzon et al., 2015*) and the International Mouse Phenotyping Consortium (IMPC) (*Bowl et al., 2017*) with gene–disease associations from literature (*Vijayakumar et al., 2019*; *Frejo et al., 2016*; *Jones and Jones, 2014*; *Gazquez and Lopez-Escamez, 2011*; *Hromatka et al., 2015*; *Oh et al., 2019*).

## Tissue isolation and immunofluorescence and 3D imaging

Temporal bones were fixed overnight at 4°C in 4% paraformaldehyde (Cat. No. 15710; Electron Microscopy Sciences; Hatfield, PA) diluted in phosphate-buffered saline (PBS; Cat. No. BP399; Thermo Fisher Scientific). Tissues were washed 3 ×~30 min in PBS. Temporal bones in some experiments were decalcified (167 mM/5% EDTA/PBS for five days at 4°C with daily exchanges), embedded in 4% agarose and sectioned at 200 μm with a vibratome. For whole mounts, ampullae were microdissected from the temporal bones. To uncover the crista sensory epithelium, the roof and wall epithelium of the ampulla were dissected away from cristae. To remove otoliths that stuck to cristae during dissection, cristae were decalcified in 167 mM (~5%) EDTA/PBS overnight at 37°C. For experiments using goat anti-Wnt3 or mouse anti-Myo7a, antigen retrieval was performed in sodium citrate buffer (10 mM Sodium citrate, 0.05% Tween 20, pH 6.0) for 45 min in a vegetable steamer. Cristae and vibratome sections were blocked (10% donkey serum/0.5% Triton-X-100/PBS) overnight at 4°C. Tissues were then incubated overnight in primary antibodies (*Table 1*) diluted in block solution at room temperature. After 3 × 1 hr washes in 0.5% Triton-X-100/PBS, tissues were incubated at least 1 hr in block, then overnight in secondary antibodies diluted in block at room temperature. After 3 × 1 hr washes in 0.5% Triton-X-100/PBS, tissues were mounted (Fluoromount-G; Cat. No. BP399; Southern Biotech; Birmingham, AL) under No. one cover slips and imaged using a Zeiss LSM880

**Table 1.** Antibodies used for immunofluorescence.

| Antigen | Host | Dilution | Cat. no. | Manufacturer | RRID |
|---|---|---|---|---|---|
| Atoh1 | Rabbit | 1:1000 | 21215–1-AP | Proteintech | AB_10733126 |
| Hsp70 | Rabbit | 1:1000 | PA5-28003 | Invitrogen | AB_2545479 |
| Iba1 | Rabbit | 1:1000 | 019–19741 | Wako | AB_839504 |
| Id1 | Rabbit | 1:1000 | BCH-1/37–2 | Biocheck | AB_2713996 |
| Myo7a | Rabbit | 1:1000 | 25–6790 | Proteus | AB_10015251 |
| Myo7a* | Mouse | 5 µg/ml | MYO7A 138–1 | DSHB | AB_2282417 |
| Nefm | Chick | 1:200 | ab134458 | Abcam | AB_2860025 |
| Ocm | Rabbit | 1:1000 | Omg4 | Swant | AB_10000346 |
| Pecam1 | Rat | 1:50 | 551262 | BD Pharmingen | AB_398497 |
| Sox2 | Goat | 1:200 | sc-17320 | Santa Cruz | AB_2286684 |
| Wnt3* | Goat | 1:125 | PA5-18516 | Thermo | AB_10979520 |

*Requires antigen retrieval.

with Airyscan. Linear adjustments to micrographs were made using 'Levels' in Adobe Photoshop v21.1.1 to increase signal:noise.

## RNA in situ hybridization

Digoxigenin (DIG)-labeled *Wnt3* probe spanning 62–1504 bp of the cDNA (NM_009521.2) was prepared according to the manufacturer's manual for DIG-11-UTP (Cat. No. 11209256910; Sigma) and the hybridization was performed on paraffin sections of inner ear as previously described (*Hayashi et al., 2007*) with differences in washes post-hybridization. Here, the following washes were performed at 65˚C after the hybridization: 60 min in 0.5X SSC/50% formamide/1% SDS, 90 min in 0.5X SSC/50% formamide/0.1% Tween 20 and 2 × 30 min washes in 0.5X SSC/0.1% Tween 20. ISH was then visualized using alkaline phosphatase-conjugated sheep anti-DIG Fab fragments (Cat. No. 11093274910; Sigma) and NBT/BCIP (Cat. No. B1911; Sigma). The RNA ISH images shown in the figure supplements are from the Allen Developing Mouse Brain Atlas (*Miller et al., 2014*). *Supplementary file 2* lists the URLs of the source images.

## Morphometric quantitation and manual cell counting

For support cell and hair cell counts, total nuclei expressing specific markers were manually counted using Fiji/ImageJ v2 (*Schindelin et al., 2012*) as described previously (*Wilkerson et al., 2018*). Total hair cell counts made in P0 B6 cristae (*Figure 2*) are compared to our previously published >P14 B6 dataset (*Wilkerson et al., 2018*). For hair cell counts, anterior and posterior cristae were grouped together and counts represent total hair cell nuclei per crista. For pigmented cell area measurements in Fiji, 'Threshold' was used to isolate the pigmented cell signal and the area above threshold measured. To test for dependence of pigmented area, $Id1^+$ support cell counts and $Atoh1^+$ cells in the support cell layer on age, two-sided Student's t-tests (alpha = 0.05) were performed using 't.test' in 'stats v3.6.2' in RStudio v1.2.5033. To test for crista hair cell count dependence on age and crista-type, two-way ANOVA and multiple linear regression analysis were performed using 'aov' and 'TukeyHSD' in 'stats v3.6.2' in RStudio v1.2.5033.

## Acknowledgements

We thank the National Institute on Deafness and Other Communications Disorders for supporting this research R01DC017126 (OBMcD), R21DC018094 (BAW) and F32DC016480 (BAW) and also the Institute for Stem Cells and Regenerative Medicine for the Innovation Pilot Award (OBMcD). We thank Dr. Cole Trapnell and Dana Jackson (UW Genome Sciences) for library preparation and sequencing. We thank members of the Bermingham-McDonogh and Reh lab for insightful discussions and Dr. Tom Reh (UW Biological Structure) for technical help with the cristae dissociations. We thank Dr. Tom Reh and Dr. Akshayalakshmi Sridhar (UW Biological Structure) for their comments on

the manuscript. We thank Dr. Henk Roelink (UC Berkeley) for the *Wnt3* plasmid. We thank Filippo Artoni and Colby Lea for their contributions.

## Additional information

### Funding

| Funder | Grant reference number | Author |
| --- | --- | --- |
| National Institutes of Health | R01DC017126 | Olivia Bermingham-McDonogh |
| National Institutes of Health | R21DC018094 | Brent A Wilkerson |
| National Institutes of Health | F32DC016480 | Brent A Wilkerson |

The funders had no role in study design, data collection and interpretation, or the decision to submit the work for publication.

### Author contributions

Brent A Wilkerson, Data curation, Formal analysis, Supervision, Validation, Investigation, Visualization, Methodology, Writing - original draft, Writing - review and editing; Heather L Zebroski, Data curation, Investigation, Methodology, Writing - review and editing; Connor R Finkbeiner, Data curation, Software, Formal analysis, Visualization, Writing - review and editing; Alex D Chitsazan, Software, Formal analysis, Visualization, Methodology, Writing - review and editing; Kylie E Beach, Data curation, Validation, Investigation, Writing - review and editing; Nilasha Sen, Renee C Zhang, Validation, Investigation; Olivia Bermingham-McDonogh, Conceptualization, Resources, Data curation, Formal analysis, Supervision, Funding acquisition, Validation, Investigation, Methodology, Writing - original draft, Project administration, Writing - review and editing

### Author ORCIDs

Brent A Wilkerson (ID) https://orcid.org/0000-0002-3532-3075
Olivia Bermingham-McDonogh (ID) https://orcid.org/0000-0002-2559-4218

### Ethics

Animal experimentation: All animal work was approved by the Institutional Animal Care and Use committee under protocol number 3123-01 and conforms to the recommendations of the NIH for animal care and use. Every effort was made to minimize any distress to the mice. Euthanasia of mice was in accordance with the AVMA guidelines.

### Decision letter and Author response

Decision letter https://doi.org/10.7554/eLife.60108.sa1
Author response https://doi.org/10.7554/eLife.60108.sa2

## Additional files

### Supplementary files

• Supplementary file 1. Cell numbers, sequencing depth, and gene detection in the crista ampullaris dataset.

• Supplementary file 2. RNA ISH images from the Allen Developing Mouse Brain Atlas.

• Transparent reporting form

### Data availability

All RNA-Seq data has been ave been deposited in NCBI's Gene Expression Omnibus (Edgar et al., 2002) and are accessible through GEO Series accession number# GSE168901.

The following dataset was generated:

| Author(s) | Year | Dataset title | Dataset URL | Database and Identifier |
|---|---|---|---|---|
| Wilkerson BA, Bermingham-McDonogh O | 2021 | Single-cell Transcriptomic Analysis of the Mouse Crista Ampullaris | https://www.ncbi.nlm.nih.gov/geo/query/acc.cgi?acc=GSE168901 | NCBI Gene Expression Omnibus, GSE168901 |

The following previously published dataset was used:

| Author(s) | Year | Dataset title | Dataset URL | Database and Identifier |
|---|---|---|---|---|
| Kelly MC | 2020 | Characterization of cochlear development at the single cell level | https://www.ncbi.nlm.nih.gov/geo/query/acc.cgi?acc=GSE137299 | NCBI Gene Expression Omnibus, GSE137299 |

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

## Appendix 1

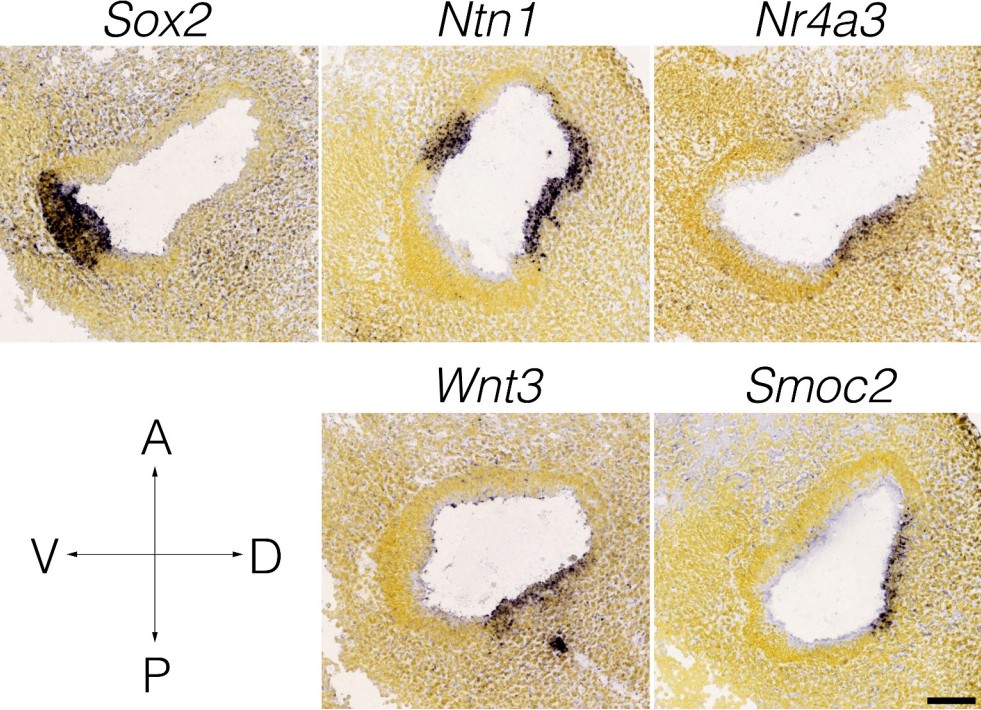

**Appendix 1—figure 1.** Wnt3 expression in the E11.5 otocyst. Panels show RNA in situ hybridization data from the Allen Developing Mouse Brain Atlas for cluster-specific markers in E15.5 B6 cristae. Note that the expression of *Wnt3* coincides with *Nr4a3/Nor1* and *Netrin1* in the presumptive canal fusion plate of the dorsolateral otocyst.

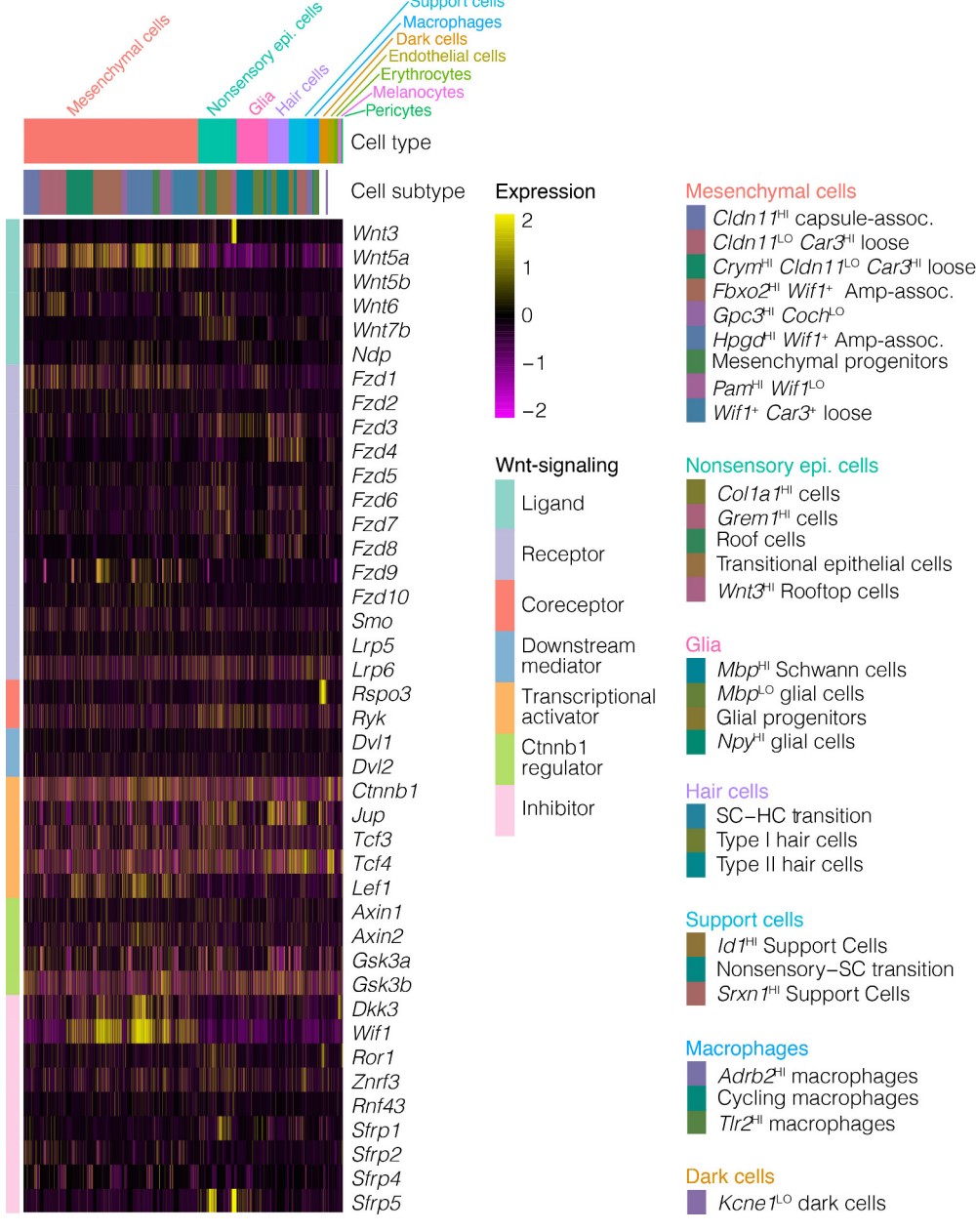

**Appendix 1—figure 2.** Wnt pathway in the perinatal crista. The expression of Wnt ligands, receptors, inhibitors and mediators is visualized by heatmap.

