## [Decision Letter]

**Acceptance summary:**

This manuscript describes a single cell transcriptomic (scRNA-seq) analysis of the mouse crista performed at different embryonic and postnatal ages. The authors identify many epithelial and mesenchymal cell types known to be present in the cristae and surrounding tissue, and identify new markers for sensory and non-sensory components of the cristae. They also perform trajectory analysis on the sensory cells to demonstrate the presence of a transient transitional cell capable of giving rise to either supporting cells or type II hair cells. In addition, they demonstrate that many genes associated with vestibular disorders in humans and mice are present in their datasets.

**Decision letter after peer review:**

Thank you for submitting your article "Novel Cell Types and Developmental Lineages Revealed by Single-cell RNA-seq Analysis of the Mouse Crista Ampullaris" for consideration by *eLife*. Your article has been reviewed by 3 peer reviewers, and the evaluation has been overseen by a Reviewing Editor and Marianne Bronner as the Senior Editor. The reviewers have opted to remain anonymous.

The reviewers have discussed the reviews with one another and the Reviewing Editor has drafted this decision to help you prepare a revised submission.

Summary:

This manuscript describes a scRNA-seq analysis of the mouse crista performed at three ages – E18, P3 and P7. The authors identify many epithelial and mesenchymal cell types known to be present in the cristae and surrounding tissue, and identify new markers for sensory and non-sensory components of the cristae. They also perform trajectory analysis on the sensory cells to demonstrate the presence of a transient transitional cell capable of giving rise to either supporting cells or type II hair cells. At the end of the paper, they demonstrate that many genes associated with vestibular disorders in humans and mice are present in their data sets.

Essential revisions:

1. After discussion between editors and reviewers, it was felt that the manuscript would fit best into the 'Tools and Resources' category. We would therefore be prepared to consider a revision if submitted for this category.

2. Two reviewers have asked for an additional developmental stage (earlier and later). After discussion, it was agreed that although both would be advantageous, an earlier stage was likely to be technically easier to accomplish. It would also help to have a clearer biological justification of the stages chosen.

3. Please provide some higher resolution validation of the genes of interest (e.g. antibody staining, RNAscope, smFISH), rather than relying on the Allen Institute data. This is particularly important for genes that identify hair cells or supporting cells in the sensory epithelium, together with some of the novel markers that have been identified. The reviewers felt that the resolution and specificity in the images (e.g. Figure 2) was not always of sufficiently high quality. It was also felt that although the data provided by the Allen Institute are an excellent resource as an initial reference, it is important that they are independently verified in a study such as this, especially given their focus on the CNS rather than the periphery.

*Reviewer #1:*

This is an extremely important paper, with lots of excellent data. The focus of the inner ear developmental field has been on the cochlea and the utricle, the former obviously because of hopes for hearing restoration and the utricle because it is an easier preparation than the crista for electrophysiologists. The current paper has made a great leap forward in focusing on the crista ampullaris, which has neither of the attributes of the former two, but instead allows us to characterise the vestibular sensory epithelium and its now-extensive cast of supporting and non-sensory epithelial cells (thanks to this paper!). I believe that this paper will go further than previous work toward an understanding of hair cell development and regeneration, but to less emphasis on ion channels (sorry, e-phys people!) and more on actual developmental factors. I would have liked to see another, more mature stage, for the RNAseq, since P7 is still somewhat immature (not all calyx endings have developed, so not all type I hair cells are present in mature form and the melanocyte area still appears to be expanding), but I do understand the limits of time and resources.

With all that said, I do have very some specific comments that I hope will improve this excellent paper. I have focused on the Figures because that is where the main results are, with less emphasis on the text. There are some inconsistencies between the figures, which I have tried to point out. Also, the logic behind what to include and in what order is not always clear. I have also tried to place this paper within the context of a couple of other important recent developmental papers on the vestibular periphery.

1. P. 4, lines 164-165. The trajectory analysis is very interesting. Does your trajectory analysis allow you to look in 3-dimensional space, rather than just a 2D UMAP? The reason that I am asking is because your data indicate that type IIs are an intermediate step between transition cells and type I hair cells, whereas the recent paper by McInturff et al. (2018), indicates that type I hair cells develop earlier than type IIs.

2. P. 5, line 195. I don't see any of these genes listed in Figure 1c. Recommend taking out this reference to Figure 1c. Also, Erg2 is the HGNC-approved gene symbol (not Krox2), thus I recommend taking out Krox2, which is just confusing, since it's not listed in Figure 3.

3. P.5, line 196. Here on the other hand is a teaser about "glial progenitors express G2/M markers", without giving any specific examples. Please include at least a couple of glial progenitor gene names. Further on in this paragraph, the authors talk about MBPlo without specifying the MBPhi population, which appear to be the Schwann cells. I think that they should put MBPhi in parentheses to make it clearer, both in the text and in the figure.

4. P. 6, lines 235-236. With regard to the statement "As a proportion of total mesenchymal cell numbers, melanocytes and dense mesenchymal cells near the boney otic capsule increase by P7 (>4-fold and >2-fold from E18, respectively)", I'm not sure that Figure 6b actually shows this. The 4-fold increase, yes, but not a greater than 2-fold increase for the dense mesenchymal cells. The figure refers only to melanocytes as a proportion of mesenchymal cells. I'm also not sure that Figure 6b shows that "proportions of loose mesenchymal cells decreased over the same developmental period (Figure 5e, Figure 6b)." I would remove Figure 6b from here and just stick with Figure 5e. Similarly, the area that the melanocytes increase is more like 3.5-fold than 4-fold.

5. P. 6, lines 249-263 and Figure 8. This paragraph reports on "Expression of genes associated with vestibular dysfunction and disease". Given the importance of these genes, I think it would be good to include the genes coded for vestibular dysfunction and disease in the heat maps on the other figures.

Figures

6. Figure 1b. Again, did the authors find any evidence of a separate lineage for type I vs type II hair cells, e.g., in a 3-dimensional, rather than 2-dimensional UMAP space? One wonders if the "Transition" cells seen in the present study correspond to the "immature hair cells", described by Rüsch et al., 1998, which have characteristics of both hair cells and supporting cells.

7. Figure 1c. Spp1 has been suggested as a marker for mature type I hair cells and Mapt as a type II marker (McInturff et al. 2018), yet the authors make no comment on the first gene and did not search for (or did not find?) the second.

8. Figure 1f. What do the authors think is the reason for there being proportionally more type I hair cells at P3 than at P7? Do some of these type I HCs die off, or is there a proportionally greater expansion of type II HCs? Or does this support what McInturff and colleagues indicate, that "over 98% of all Type I hair cells develop prior to birth while over 98% of Type II hair cells develop post-natally"? In the adults of many rodent species, there seem to be slightly more type I hair cells than type II, so the results here are a bit surprising.

9. Figure 1i. The authors have lumped together the anterior and posterior cristae. Is this because they could not tell them apart once they were dissected out? Are the # of hair cells grouped values or per individual crista? Do the AC and PC each have more cells than the HC? – this seems unlikely since they both have a nonsensory part, the eminentia cruciatum, in the middle. If that's not the case and the numbers are pooled for the vertical cristae, then they would seem to each have many fewer hair cells than the HC (on average ~700 each, compared to ~1100?). Furthermore, the number of hair cells still appears to be growing between P14-21 and 6 months, as shown in Figure 1i, another argument for including a later stage for the RNAseq.

10. Figure 4. The authors have described the localization of Aldh1a3 in the transitional epithelium. This gene and its complementary partner, Cyp26b1, are both related to the action of the important developmental morphogen, retinoic acid, and have recently been described (Ono et al. 2020) as being important for the differentiation of particular subregions of the sensory epithelium, such as central and peripheral crista subregions and the striolar and extrastriolar subregions in the otolith organs. Since it is one of the genes found by the authors in the RNAseq database, and even though it is associated with "Erythrocytes" and "Loose ampulla-proximal mesenchyme 2", I would still like to see Cyp26b1 listed, perhaps appropriately in Figure 5. Can the authors add this gene?

11. Figures 4 and 5. I am delighted by the discovery of a novel set of markers demarcating the Wnt3+ rooftop domain. The authors have suggested on page 8 of the manuscript that this may be related to the cupula, however in Figure 4g, it runs perpendicular rather than parallel to the crista (note the position of the non-sensory eminentia area), so I am not sure how that would work. Perhaps there is a sub-portion of the rooftop that provides an attachment locus for the cupula? And by the way, the brackets and arrows in Figures 4e, f, g and in Figure 5a, b are too small. Please increase their width by at least 3 times. I like the "Marker specificity" coding (novel, known, etc.) in Figures 2 and 4. Could the authors please make a similar column for Figures 1, 3, 5 and 7? I think it would be informative and highlight the innovation of this paper.

12. Figure 5. The arrows in Figure 5b are again a bit too small. Could the authors make them a bit bigger, or better yet, add a label name for the upper arrow as they have in Figure 1-4 and 7? Again, the legend refers to "G2/M markers" without listing any.

13. Figure 6. There are 1,000 genes associated with melanocytes, yet there is no heat map for this portion of the Results. Did the authors find no difference in expression patterns (did they all turn on at once, as suggested by Figure 1c?)

14. Figure 7. I think that the "Endothelial cells" and "Pericytes" clusters in panel a would benefit from an additional arrow to point out these tiny clusters. Are any of these genes "novel" for these categories? There is no legend for the cell type colour-coding.

15. Figure 8: The colour coding between type I and type II hair cells is too close (and I'm not even colour-blind), so it is difficult to distinguish them in both the legend and on the heat map. Also, I like that there is a column relating these genes to other classifications, such as IMPC, balance and vestibular disorders. Given these important classifications, can the authors be sure to include the genes listed under these categories, e.g., balance, disorders, development, etc. in the heat maps of their respective categories?

*Reviewer #2:*

Wilkerson et al. used 10x genomic single cell analysis approach to study mouse crista ampullaris at three different ages E18, P3 and P7. They performed several bioinformatic analyses and identified expected cell types such as hair cells and support cells, glial cells and several new non-sensory cell types. Basically, the bioinformatic analysis methods are appropriate, as the previous known ampullar specific genes could be enriched after analysis, for example the Zpld1 gene (in the paper, the author used Zpld instead, and it should be a typo). I have 3 main comments.

1. The current content of the manuscript was well written and understood, but for research article in *eLife*, in my own point-of-view, it is mainly descriptive so far and a deeper biological insights should be provided. Indeed, there are several good points that could be further explored to strengthen the paper. For example, the cells with high Id1 expression are of particular interest to understand ampullar development. It will be a good plus if the authors could perform in vivo fate mapping analysis using the Id1-CreER mouse line (Cell Stem Cell. 2009 Nov 6; 5(5): 515-526) at multiple ages. This experiment will determine whether the hypothesis is correct that Cldn4+ cells are progeny of Id1+ cells.

Although the authors named them as support cells, they should be referred as progenitor cells. Instead, the Cldn4+ cells are differentiated support cells.

2. The current three ages (E18, P3 and P7 ) are not complete to cover the main developmental program of ampullar ogran. Because vestibular ogran differentiation occurs earlier than cochlea, samples from E12-E13 age should be included for a more complete story.

3. The RNA in situ data that were used to validate single cell RNA-Seq data were totally from Allen Developing Mouse Brain Atlas. It should be generated in mouse samples at E18, P3 and P7 by the authors for better in situ validation. In addition, the resolution of those ISH images could be better. I have difficulty in judging the cell types precisely.

*Reviewer #3:*

The paper is technically unimpeachable and the authors have used Allen Brain Atlas data to help validate some of their different genes. The paper does not offer any significant biological or conceptual advance, but its data set will be of interest to those studying vestibular development.

My only comment of significance is that the authors should comment on the ages chosen and discuss when mouse cristae are believed to be fully mature – or at least, when hair cell addition ceases. This might help place their trajectory analysis on a more biological footing.

[Editors' note: further revisions were suggested prior to acceptance, as described below.]

Thank you for resubmitting your work entitled "Novel Cell Types and Developmental Lineages Revealed by Single-cell RNA-seq Analysis of the Mouse Crista Ampullaris" for further consideration by *eLife*. Your revised article has been evaluated by Marianne Bronner as the Senior Editor, Tanya Whitfield as the Reviewing Editor, and an anonymous reviewer.

The manuscript has been improved and we find that you have addressed the essential revisions. However, one of the reviewers points out that there are still areas that need further clarification (see full review below). We recommend that you address these issues where possible, adding further clarification and explanation where requested. We are not requesting any further experimental work.

*Reviewer #2:*

In the revised manuscript, the authors included an additional age (E16) and combined it with previous E18, P3 and P7 ages. So far, the manuscript is still mainly descriptive and limited biological insights are present. While I appreciate the deep analysis of the data, tool and resources format is recommended, as many proposed ideas do need future in vivo experiments to validate.

My main concerns or the main weakness of the manuscript are still the general not high quality of images, compared to others previously published in *eLife*, as well as the limited gene numbers detected in E16 samples. Furthermore, a better cartoon should be provided to clearly tell readers what cells are included. Until reading the rebuttal letters, I did not know that the ampullar ganglions were not included during dissection. Therefore, the glial cells, which are included in this manuscript, should be only a small portion close to neural fibers and should not represent the majority of the glial cells that intermingle with vestibular neuron bodies. It should be clearly described, otherwise will be a misleading. Please see other details below.

1. For the pseudotime or trajectory analysis, it is better to annotate/label the cells of their real developmental ages (in this study, there are 4 ages). In particular, for ampullar hair cells and supporting cells, such analysis is necessary. It is used in many previous single cell RNA-seq studies. One advantage of this is to further validate to what degree the pseudotime of one cell matches its real developmental ages. If the pseudotime contradicts with real developmental age of one cell, how to interpret the data? It is likely the case in ampullar organ, as HCs and SCs are continuously added between E16 and P7. In other words, I do recommend HCs and SCs need more thorough analysis.

2. The newly added E16 data, compared to other ages, generally have significantly lower gene numbers detected per cell, according to the 'Supplementary file 1-cell numbers '. It is unexpected because normally cells at younger ages are easier to get good cell suspensions for 10x analysis and more genes would be detected. The authors should not ignore this issue, and instead should explain the details. At least, the readers can keep in mind this issue while reading the manuscript.

3. Experiments regarding Slc1a3-CreER mouse strain, which is added in the revised version, is not described/included in result or method section at all. Thus, it is not clear when tamoxifen (dosage) is injected and tissues are analyzed. Please provide detailed information. Indeed, it is a good opportunity for authors to thoroughly characterize this line to provide high-resolution images with good qualities (single cell level visualization), as many images did not match my personal standard toward *eLife* papers. Both section and whole mount images are needed to help readers to appreciate the expression pattern of Slc1a3.

---

## [Author Response]

Essential revisions:1. After discussion between editors and reviewers, it was felt that the manuscript would fit best into the 'Tools and Resources' category. We would therefore be prepared to consider a revision if submitted for this category.

We agree to the Tools and Resources” category although I should note that we have now included a more in depth analysis of the data.

2. Two reviewers have asked for an additional developmental stage (earlier and later). After discussion, it was agreed that although both would be advantageous, an earlier stage was likely to be technically easier to accomplish. It would also help to have a clearer biological justification of the stages chosen.

We have added an additional age (E16) and reanalyzed the aggregate data.

3. Please provide some higher resolution validation of the genes of interest (e.g. antibody staining, RNAscope, smFISH), rather than relying on the Allen Institute data. This is particularly important for genes that identify hair cells or supporting cells in the sensory epithelium, together with some of the novel markers that have been identified. The reviewers felt that the resolution and specificity in the images (e.g. Figure 2) was not always of sufficiently high quality. It was also felt that although the data provided by the Allen Institute are an excellent resource as an initial reference, it is important that they are independently verified in a study such as this, especially given their focus on the CNS rather than the periphery.

We have added additional validations and included higher mag images.

Reviewer #1:This is an extremely important paper, with lots of excellent data. The focus of the inner ear developmental field has been on the cochlea and the utricle, the former obviously because of hopes for hearing restoration and the utricle because it is an easier preparation than the crista for electrophysiologists. The current paper has made a great leap forward in focusing on the crista ampullaris, which has neither of the attributes of the former two, but instead allows us to characterise the vestibular sensory epithelium and its now-extensive cast of supporting and non-sensory epithelial cells (thanks to this paper!). I believe that this paper will go further than previous work toward an understanding of hair cell development and regeneration, but to less emphasis on ion channels (sorry, e-phys people!) and more on actual developmental factors. I would have liked to see another, more mature stage, for the RNAseq, since P7 is still somewhat immature (not all calyx endings have developed, so not all type I hair cells are present in mature form and the melanocyte area still appears to be expanding), but I do understand the limits of time and resources.With all that said, I do have very some specific comments that I hope will improve this excellent paper. I have focused on the Figures because that is where the main results are, with less emphasis on the text. There are some inconsistencies between the figures, which I have tried to point out. Also, the logic behind what to include and in what order is not always clear. I have also tried to place this paper within the context of a couple of other important recent developmental papers on the vestibular periphery.1. P. 4, lines 164-165. The trajectory analysis is very interesting. Does your trajectory analysis allow you to look in 3-dimensional space, rather than just a 2D UMAP? The reason that I am asking is because your data indicate that type IIs are an intermediate step between transition cells and type I hair cells, whereas the recent paper by McInturff et al. (2018), indicates that type I hair cells develop earlier than type IIs.

We have added an E16 sample to the data for analysis. This sample actually nicely filled out the dataset and now allows us to see the common precursor to Type I and Type II hair cells. We have also identified specific markers for the transition to either Type I (Sox21) or Type II hair cells (Insm1). In the earlier version of the manuscript we did not have enough cells of the appropriate age to show the branch point from common precursor to hair cell. We also include a supplemental figure (Figure 2S3) to show RNA velocity that clearly indicates velocity evident in the transition to Type II hair cells which is probably because of the fact that most of the Type I hair cells are already differentiated. In contrast to what is seen in the utricle (McInturff, 2018: Burns, 2015) we see Anxa4 positive hair cells in the E16 sample for crista suggesting that Type II hair cells are differentiating prenatally in this organ.

2. P. 5, line 195. I don't see any of these genes listed in Figure 1c. Recommend taking out this reference to Figure 1c. Also, Erg2 is the HGNC-approved gene symbol (not Krox2), thus I recommend taking out Krox2, which is just confusing, since it's not listed in Figure 3.

Krox20 and the 1c reference have been removed and now we refer to Egr2.

3. P.5, line 196. Here on the other hand is a teaser about "glial progenitors express G2/M markers", without giving any specific examples. Please include at least a couple of glial progenitor gene names. Further on in this paragraph, the authors talk about MBPlo without specifying the MBPhi population, which appear to be the Schwann cells. I think that they should put MBPhi in parentheses to make it clearer, both in the text and in the figure.

Examples of G2/M markers appearing in the heatmap are listed in the revised text. G2/M markers are labeled as such in the revised Figure 3. Text and labels now indicate that Schwann cells are *Mbp*^HI^.

4. P. 6, lines 235-236. With regard to the statement "As a proportion of total mesenchymal cell numbers, melanocytes and dense mesenchymal cells near the boney otic capsule increase by P7 (>4-fold and >2-fold from E18, respectively)", I'm not sure that Figure 6b actually shows this. The 4-fold increase, yes, but not a greater than 2-fold increase for the dense mesenchymal cells. The figure refers only to melanocytes as a proportion of mesenchymal cells. I'm also not sure that Figure 6b shows that "proportions of loose mesenchymal cells decreased over the same developmental period (Figure 5e, Figure 6b)." I would remove Figure 6b from here and just stick with Figure 5e. Similarly, the area that the melanocytes increase is more like 3.5-fold than 4-fold.

Figure references and text in lines 235-236 were revised for clarity per Reviewer 1’s suggestions, lines 194-196 in current version.

“As a proportion of total mesenchymal cell numbers, melanocytes increase ~3.5-fold by P7 from E18 (Figure 6b). Similarly, the area occupied by pigmented cells increases ~3.5-fold in P7 versus P0 crista ampullaris (Figure 6c-e). “

5. P. 6, lines 249-263 and Figure 8. This paragraph reports on "Expression of genes associated with vestibular dysfunction and disease". Given the importance of these genes, I think it would be good to include the genes coded for vestibular dysfunction and disease in the heat maps on the other figures.

We are showing all of the vestibular dysfunction genes in the heat map as a Supplementary file (because of its size this file (Figure 9S1) is best viewed with a document viewer) with a subset of these genes shown in Figure 9. Throughout the manuscript genes associated with vestibular dysfunction are indicated by a color bar.

Figures6. Figure 1b. Again, did the authors find any evidence of a separate lineage for type I vs type II hair cells, e.g., in a 3-dimensional, rather than 2-dimensional UMAP space? One wonders if the "Transition" cells seen in the present study correspond to the "immature hair cells", described by Rüsch et al., 1998, which have characteristics of both hair cells and supporting cells.

We have now added an additional age E16 and this nicely fills out the data such that we do indeed see the transitions to both Type I and Type II hair cells. It appears that they take different trajectories from a common progenitor. Indeed, we have now identified specific markers of these transitions. Text has been added in the Discussion, to indicate that we think they represent immature hair cells as described in Rüsch et al. 1998 and this reference has been added. Thank you for pointing out the omission.

7. Figure 1c. Spp1 has been suggested as a marker for mature type I hair cells and Mapt as a type II marker (McInturff et al. 2018), yet the authors make no comment on the first gene and did not search for (or did not find?) the second.

The revised heatmap shows that Spp1 and other hair cell subtype-specific markers identified by McInturff are largely specific in crista also. We found Mapt only in only 4 Type II hair cells and none in Type I. We didn’t include this because of the sparse detection which is a limitation of 10X. We have also identified other markers for both hair cell types in the crista.

8. Figure 1f. What do the authors think is the reason for there being proportionally more type I hair cells at P3 than at P7? Do some of these type I HCs die off, or is there a proportionally greater expansion of type II HCs? Or does this support what McInturff and colleagues indicate, that "over 98% of all Type I hair cells develop prior to birth while over 98% of Type II hair cells develop post-natally"? In the adults of many rodent species, there seem to be slightly more type I hair cells than type II, so the results here are a bit surprising.

Postnatal expansion of type II hair cells in the utricle is described in Rüsch et al. 1998, Burns et al. 2012 and McInturff et al. 2018. Consistent with this, we found evidence in crista for postnatal expansion of both type I and type II hair cells, with more of the latter. We believe that this result is more consistent with the Rusch paper, which also shows expansion of type I at a lower level postnatally. With the addition of the new data and reclustering we no longer see a decrease in type I from P3 to P7.

9. Figure 1i. The authors have lumped together the anterior and posterior cristae. Is this because they could not tell them apart once they were dissected out? Are the # of hair cells grouped values or per individual crista? Do the AC and PC each have more cells than the HC? – this seems unlikely since they both have a nonsensory part, the eminentia cruciatum, in the middle. If that's not the case and the numbers are pooled for the vertical cristae, then they would seem to each have many fewer hair cells than the HC (on average ~700 each, compared to ~1100?). Furthermore, the number of hair cells still appears to be growing between P14-21 and 6 months, as shown in Figure 1i, another argument for including a later stage for the RNAseq.

Details on the counting method have been added in Methods to clarify that total hair cells per crista and that anterior cristae (AC) and posterior cristae (PC) are grouped together. Although AC and PC have the eminentia cruciatum, horizontal cristae have significantly smaller sensory patches than anterior and posterior cristae in B6 and CBA mice (Wilkerson et al. 2018). Hair cell density in AC/PC and HC are similar if the eminentia cruciatum is excluded from AC/PC area (Wilkerson et al. 2018), however because of the smaller sensory area in the horizontal cristae there are fewer hair cells than found in the AC or PC. We find no statistically significant difference in hair cell numbers in P14-21 cristae versus those at 3 months or at 6 months.

10. Figure 4. The authors have described the localization of Aldh1a3 in the transitional epithelium. This gene and its complementary partner, Cyp26b1, are both related to the action of the important developmental morphogen, retinoic acid, and have recently been described (Ono et al. 2020) as being important for the differentiation of particular subregions of the sensory epithelium, such as central and peripheral crista subregions and the striolar and extrastriolar subregions in the otolith organs. Since it is one of the genes found by the authors in the RNAseq database, and even though it is associated with "Erythrocytes" and "Loose ampulla-proximal mesenchyme 2", I would still like to see Cyp26b1 listed, perhaps appropriately in Figure 5. Can the authors add this gene?

Cyp26b1 is expressed sparsely in many of the cell types of the crista, including sensory epithelial cells. It is most highly expressed in the mesenchyme and now is added to the map of Figure 5.

11. Figures 4 and 5. I am delighted by the discovery of a novel set of markers demarcating the Wnt3+ rooftop domain. The authors have suggested on page 8 of the manuscript that this may be related to the cupula, however in Figure 4g, it runs perpendicular rather than parallel to the crista (note the position of the non-sensory eminentia area), so I am not sure how that would work. Perhaps there is a sub-portion of the rooftop that provides an attachment locus for the cupula? And by the way, the brackets and arrows in Figures 4e, f, g and in Figure 5a, b are too small. Please increase their width by at least 3 times. I like the "Marker specificity" coding (novel, known, etc.) in Figures 2 and 4. Could the authors please make a similar column for Figures 1, 3, 5 and 7? I think it would be informative and highlight the innovation of this paper.

The cupula’s attachment point in the rooftop is described in papers from the Rabbit lab but the difference in orientation of the cupula and the Wnt3 domain is a good point that and the fact that the Wnt3 continues this restricted expression pattern along the canal caused us to reconsider our speculation about the Wnt3 domain as a cupula attachment point and we decided to remove the related text from the Discussion.

We increased the size of the brackets and arrows in Figure 4 and 5. We added the known/novel annotations to the other heatmaps.

12. Figure 5. The arrows in Figure 5b are again a bit too small. Could the authors make them a bit bigger, or better yet, add a label name for the upper arrow as they have in Figure 1-4 and 7? Again, the legend refers to "G2/M markers" without listing any.

Arrows in Figure 5 have been enlarged and labeled. G2/M markers are now listed in the text.

13. Figure 6. There are 1,000 genes associated with melanocytes, yet there is no heat map for this portion of the Results. Did the authors find no difference in expression patterns (did they all turn on at once, as suggested by Figure 1c?)

We performed differential expression analysis across developmental timepoints and included a heatmap showing significantly altered genes. This is in revised Figure 6.

14. Figure 7. I think that the "Endothelial cells" and "Pericytes" clusters in panel a would benefit from an additional arrow to point out these tiny clusters. Are any of these genes "novel" for these categories? There is no legend for the cell type colour-coding.

With the addition of the E16 data the clusters of endothelial cells and pericytes are easier to see. No novel markers were found for these cell types.

15. Figure 8: The colour coding between type I and type II hair cells is too close (and I'm not even colour-blind), so it is difficult to distinguish them in both the legend and on the heat map. Also, I like that there is a column relating these genes to other classifications, such as IMPC, balance and vestibular disorders. Given these important classifications, can the authors be sure to include the genes listed under these categories, e.g., balance, disorders, development, etc. in the heat maps of their respective categories?

We changed the colors to better contrast the type I and type II hair cell clusters. Genes implicated in vestibular dysfunction and disease have been identified by color bars in all Figures.

Reviewer #2:Wilkerson et al. used 10x genomic single cell analysis approach to study mouse crista ampullaris at three different ages E18, P3 and P7. They performed several bioinformatic analyses and identified expected cell types such as hair cells and support cells, glial cells and several new non-sensory cell types. Basically, the bioinformatic analysis methods are appropriate, as the previous known ampullar specific genes could be enriched after analysis, for example the Zpld1 gene (in the paper, the author used Zpld instead, and it should be a typo).

We fixed the typo. Thank you.

I have 3 main comments.1. The current content of the manuscript was well written and understood, but for research article in eLife, in my own point-of-view, it is mainly descriptive so far and a deeper biological insights should be provided. Indeed, there are several good points that could be further explored to strengthen the paper. For example, the cells with high Id1 expression are of particular interest to understand ampullar development. It will be a good plus if the authors could perform in vivo fate mapping analysis using the Id1-CreER mouse line (Cell Stem Cell. 2009 Nov 6; 5(5): 515-526) at multiple ages. This experiment will determine whether the hypothesis is correct that Cldn4+ cells are progeny of Id1+ cells.Although the authors named them as support cells, they should be referred as progenitor cells. Instead, the Cldn4+ cells are differentiated support cells.

The Reviewing Editor agrees with your point and so we have resubmitted the manuscript as a Tools and Resources paper rather than a research article. We are excited about the possibility that Id1+ support cells are progenitors, too. We have adding text that recommends the suggested experiment for a future study in the Discussion. Because the implications of findings from this big data-driven study are too numerous for us to validate in a timely fashion (and because independent validation is best!), we are eager to publish and release the dataset as a resource for the field.

2. The current three ages (E18, P3 and P7 ) are not complete to cover the main developmental program of ampullar ogran. Because vestibular ogran differentiation occurs earlier than cochlea, samples from E12-E13 age should be included for a more complete story.

We have clarified in the Introduction that our choice to analyze the perinatal stages was made because of our own interest specifically in the developmental window of support cell competence to convert to hair cells. We have now added single cell RNA-seq data from E16 cristae. E12-13 would be interesting but very challenging to dissect. The additional E16 dataset did fill out the transitions and we remade the figures to include this data in the aggregate analyses.

3. The RNA in situ data that were used to validate single cell RNA-Seq data were totally from Allen Developing Mouse Brain Atlas. It should be generated in mouse samples at E18, P3 and P7 by the authors for better in situ validation. In addition, the resolution of those ISH images could be better. I have difficulty in judging the cell types precisely.

We increased size of in situ images in the supplement. As stated above we have added an E16 sample and this sample closely matches the E15.5 in situ data from the Allen Institute. We have validated additional markers that discriminate the novel support cell subtypes using immunofluorescence and included higher-mag images of anti-Id1, and Atoh1 and added a new supplemental figure showing Ocm and neurofilament staining (Figure 2S1).

Reviewer #3:The paper is technically unimpeachable and the authors have used Allen Brain Atlas data to help validate some of their different genes. The paper does not offer any significant biological or conceptual advance, but its data set will be of interest to those studying vestibular development.My only comment of significance is that the authors should comment on the ages chosen and discuss when mouse cristae are believed to be fully mature – or at least, when hair cell addition ceases. This might help place their trajectory analysis on a more biological footing.

We have clarified in the Introduction that our focus on perinatal stages as a complement to our lab’s ongoing research on the developmental window of support cell competence to convert to hair cells. We also see no further addition of hair cells after postnatal day 14 but have no data on functional maturity.

[Editors' note: further revisions were suggested prior to acceptance, as described below.]

Reviewer #2:In the revised manuscript, the authors included an additional age (E16) and combined it with previous E18, P3 and P7 ages. So far, the manuscript is still mainly descriptive and limited biological insights are present. While I appreciate the deep analysis of the data, tool and resources format is recommended, as many proposed ideas do need future in vivo experiments to validate.My main concerns or the main weakness of the manuscript are still the general not high quality of images, compared to others previously published in eLife, as well as the limited gene numbers detected in E16 samples.

Gene numbers in the E16 were not a limitation for our analysis: we resolved all the major cell groups and subtypes found in the other samples as well as both of the lineages of the hair cell subtypes (Figure 2). With batch correction, samples with varying depth were integrated and compared effectively. Furthermore, low gene numbers are common in single nucleus and SCI-seq samples, which are also suitable for clustering, trajectory analysis, etc.

The images show that markers from single cell analysis predict localization in tissue. We added new images, Z-stack videos and URLs as supplements to the figures so that readers can evaluate markers in additional sections.

Furthermore, a better cartoon should be provided to clearly tell readers what cells are included. Until reading the rebuttal letters, I did not know that the ampullar ganglions were not included during dissection. Therefore, the glial cells, which are included in this manuscript, should be only a small portion close to neural fibers and should not represent the majority of the glial cells that intermingle with vestibular neuron bodies. It should be clearly described, otherwise will be a misleading.

We added text to the Results and Methods in the areas pertaining to the dissection and glia to clarify that the vestibular ganglion was not analyzed and to specifically state what cells are included:

“The specific tissue for dissociation into single cells included the SE, transitional epithelium, the epithelial cells making up the walls of the ampulla, ampullar associated mesenchyme and the glial cells associated with the peripheral processes of the vestibular neurons. The cell bodies of the neurons were not included.”

Please see other details below.1. For the pseudotime or trajectory analysis, it is better to annotate/label the cells of their real developmental ages (in this study, there are 4 ages). In particular, for ampullar hair cells and supporting cells, such analysis is necessary. It is used in many previous single cell RNA-seq studies. One advantage of this is to further validate to what degree the pseudotime of one cell matches its real developmental ages. If the pseudotime contradicts with real developmental age of one cell, how to interpret the data? It is likely the case in ampullar organ, as HCs and SCs are continuously added between E16 and P7. In other words, I do recommend HCs and SCs need more thorough analysis.

We added the UMAPs with cells color-coded by developmental stage as a supplement to figures showing trajectory analysis (i.e., Figures 2, 3 and 7).

Pseudotime and developmental stage correlate in synchronous developmental processes. For example, we observe higher pseudotime correlation with developmental stage in the glial and dark cell trajectories and we showed in our analysis of cell proportions that their progenitors disappear more rapidly with developmental time in the respective figures. By contrast, the hair cell trajectory shows lower correlation between pseudotime and developmental stage vs. other trajectories. We identified immature hair cells as well as transitioning hair cells at a range of pseudotimes at all stages. Immature hair cells decreased also but more slowly than the glial progenitors. We interpret this as evidence for an asynchronous process of hair cell differentiation in the cristae similar to what has been described in the utricle.

To otherwise validate the trajectories suggested by pseudotime analysis, we performed RNA velocity analysis on the epithelial cells in the dataset (Figure 2S4). RNA velocity is derived from the ratio of spliced to unspliced/nascent transcripts in adjacent cells and reflects the trajectory of cells in the UMAP transcriptional space. This analysis likewise suggests transitions between support cells and type II hair cells and also shows low velocity in the space between type II and type I hair cells. It also corroborates the transition between the nonsensory ampulla cells and support cells. Currently RNA velocity is the most accurate way to identify developmental transitions in single cell transcriptomic data to our knowledge.

2. The newly added E16 data, compared to other ages, generally have significantly lower gene numbers detected per cell, according to the 'Supplementary file 1-cell numbers '. It is unexpected because normally cells at younger ages are easier to get good cell suspensions for 10x analysis and more genes would be detected. The authors should not ignore this issue, and instead should explain the details. At least, the readers can keep in mind this issue while reading the manuscript.

The gene numbers in the E16 are adequate to identify major cell types and to explore subtypes as well as developmental trajectories.

In this study, cell yields were higher at later stages. Epithelial cell proportions including hair cells were greatest in the E16 (Supp. file 3 and Figure 1b), suggesting that epithelial cell types might be easier to isolate earlier possibly because of easier dissociation. We observed higher mesenchymal cell proportions in the later samples and we speculate that is because mesenchyme dissociates more readily than epithelial cells with mature junctions. According to the 10X manufacturers sometimes the number of genes per cell is lower in a more homogenous tissue so perhaps this reflects a lack of full differentiation of the epithelium at E16.

3. Experiments regarding Slc1a3-CreER mouse strain, which is added in the revised version, is not described/included in result or method section at all. Thus, it is not clear when tamoxifen (dosage) is injected and tissues are analyzed. Please provide detailed information. Indeed, it is a good opportunity for authors to thoroughly characterize this line to provide high-resolution images with good qualities (single cell level visualization), as many images did not match my personal standard toward eLife papers. Both section and whole mount images are needed to help readers to appreciate the expression pattern of Slc1a3.

We added details to the Figure Legend specifying that ears were collected opportunistically from mice used in the cited study by our colleagues, as well as the crosses and dosing they used. Previous studies by Stone et al. *JARO.* 2018 and Mellado Lagarde et al. *PNAS.* 2014 characterized Slc1a3-CreER activity in the utricle and cochlea. We used this tissue to validate the single cell finding of Slc1a3 expression in the mesenchyme.